# A cryptic K48 ubiquitin chain binding site on UCH37 is required for its role in proteasomal degradation

Jiale Du[1], Sandor Babik[1], Yanfeng Li[1], Kirandeep K Deol[1†], Stephen J Eyles[2], Jasna Fejzo[3], Marco Tonelli[4], Eric Strieter[1,5*]

[1]Department of Chemistry, University of Massachusetts Amherst, Amherst, United States; [2]Mass Spectrometry Core Facility, Institute for Applied Life Sciences (IALS), University of Massachusetts Amherst, Amherst, United States; [3]Biomolecular NMR Core Facility, Institute for Applied Life Sciences (IALS), University of Massachusetts Amherst, Amherst, United States; [4]National Magnetic Resonance Facility at Madison (NMRFAM), University of Wisconsin-Madison, Madison, United States; [5]Molecular & Cellular Biology Graduate Program, University of Massachusetts Amherst, Amherst, United States

*For correspondence:
estrieter@umass.edu

Present address: †Department of Nutritional Sciences and Toxicology, University of California, Berkeley, United States

**Abstract** Degradation by the 26 S proteasome is an intricately regulated process fine tuned by the precise nature of ubiquitin modifications attached to a protein substrate. By debranching ubiquitin chains composed of K48 linkages, the proteasome-associated ubiquitin C-terminal hydrolase UCHL5/UCH37 serves as a positive regulator of protein degradation. How UCH37 achieves specificity for K48 chains is unclear. Here, we use a combination of hydrogen-deuterium mass spectrometry, chemical crosslinking, small-angle X-ray scattering, nuclear magnetic resonance (NMR), molecular docking, and targeted mutagenesis to uncover a cryptic K48 ubiquitin (Ub) chain-specific binding site on the opposite face of UCH37 relative to the canonical S1 (cS1) ubiquitin-binding site. Biochemical assays demonstrate the K48 chain-specific binding site is required for chain debranching and proteasome-mediated degradation of proteins modified with branched chains. Using quantitative proteomics, translation shutoff experiments, and linkage-specific affinity tools, we then identify specific proteins whose degradation depends on the debranching activity of UCH37. Our findings suggest that UCH37 and potentially other DUBs could use more than one S1 site to perform different biochemical functions.

## Editor's evaluation

This study identifies a previously unknown, non-canonical ubiquitin-binding site on the backside of the proteasome-associated deubiquitinase UCH37 that is responsible for the specific removal of Lys48-linked branches from ubiquitin chains. Using a broad array of biochemical and biophysical approaches, the authors characterize the ubiquitin binding modes and critical motifs of this new site, and investigate its effects on ubiquitin-dependent protein degradation by the 26S proteasome in vitro and in cells. These findings represent an important advance to our understanding of ubiquitin cleavage at the 26S proteasome and its role in regulating protein turnover by the ubiquitin-proteasome system.

## Introduction

Protein ubiquitination is a dynamic post-translational modification that has profound effects on cellular function (*Oh et al., 2018*). Once tagged with ubiquitin (Ub), a protein target can be forced to change cellular location, assemble into a multiprotein complex, or suffer the fate of degradation.

These diverse functions are orchestrated by a wide array of different Ub modifications. In addition to mono- and multimono-ubiquitination, Ub can be assembled into polymeric chains composed of different linkages (M1, K6, K11, K27, K29, K33, K48, and K63), lengths, and architectures (unbranched and branched) (*Komander and Rape, 2012*). Central to the regulation of Ub-dependent signaling events is the ability to remove Ub modifications. These actions are executed by a family of proteolytic enzymes called deubiquitinases (DUBs).

There are nearly 100 human DUBs that fall into seven distinct subfamilies: USPs, OTUs, UCHs, Josephin, MINDY, ZUP1, and JAMM/MPN+ (*Clague et al., 2019*; *Leznicki and Kulathu, 2017*; *Mevissen and Komander, 2017*). Members of the OTU, MINDY, JAMM, and ZUP1 families generally target specific linkages within Ub chains, leaving behind the Ub moiety directly attached to a substrate (*Abdul Rehman et al., 2021*; *Abdul Rehman et al., 2016*; *Hermanns et al., 2018*; *Hewings et al., 2018*; *Kwasna et al., 2018*; *Mevissen et al., 2013*). USPs, on the other hand, tend to display selectivity for the protein directly anchored to Ub and are rather promiscuous toward different chain types (*Faesen et al., 2011*; *Ritorto et al., 2014*). In some cases, there are DUBs capable of targeting a specific chain type and removing Ub directly from a variety of different substrates (*Gersch et al., 2017*). These fundamentally distinct activities are thought to require the same primary Ub-binding site (termed the S1 site), which steers the Ub C-terminal scissile bond into the active site for cleavage. A secondary binding site—a S1′ site—enables substrate discrimination. With chain type-specific DUBs, for example, the S1 site interacts with the Ub (i.e. distal Ub) whose C-terminus is attached to another Ub molecule, and the S1′ site positions the proximal Ub of a chain (i.e. the subunit supplying the Lys or N-terminus) to allow only one linkage to gain access to the catalytic cleft (*Mevissen et al., 2016*; *Mevissen et al., 2013*). We thought the same set of rules would apply to the K48 chain-specific DUB UCH37.

Through association with the 26 S proteasome and INO80 chromatin remodeling complexes, UCH37 (also called UCHL5) plays a fundamental role in many cellular processes (*Hamazaki et al., 2006*; *Jørgensen et al., 2006*; *Qiu et al., 2006*; *Yao et al., 2008*; *Yao et al., 2006*). Although its function in the context of the INO80 complex is poorly understood, we (*Deol et al., 2020*) and others (*Song et al., 2021*) recently reported that the proteasome-bound form specifically binds K48 chains and catalyzes the removal of K48 branchpoints to promote degradation. Protein substrates that depend on the Ub chain debranching activity of UCH37 for degradation, however, remain unknown. Complicating the identification of substrates is the fact that UCH37 has other known biochemical activities. As a member of the UCH family, UCH37 removes small adducts from the C-terminus of Ub and more recent studies have shown that the DUB can act as a peptidase by cleaving Ub from the N-terminus of proteins (*Bett et al., 2015*; *Davies et al., 2021*).

Based on known structures of monoUb bound to UCH37 (*Sahtoe et al., 2015*; *Vander Linden et al., 2015*), the hypothesis is that both the C-terminal hydrolase and debranching activities utilize the same S1 site during catalysis. The S1 site is composed of a series of residues that form a rim leading into the active site (*Figure 1A*). The DEUBAD domain of RPN13 further promotes Ub binding by properly positioning both the flexible active site crossover loop (CL), which is a characteristic feature of all UCHs (*Johnston et al., 1999*; *Popp et al., 2009*), and the C-terminal helical region of UCH37. One would therefore speculate that during chain debranching, the S1 site would be occupied by the K48-linked distal Ub, as this subunit is the one removed from the branched chain (*Figure 1B*). In this orientation, the other two Ub subunits at the branchpoint would also be expected to engage UCH37; however, the precise mechanism by which branched chains are selectively processed by UCH37 remains unknown. A region that has received little attention in any of the UCHs is the face located on the opposite side of the CL relative to the S1 site. We thought that by defining how K48 chains interact with UCH37, we could begin to dissect the impact of K48 chain binding and debranching on a proteome-wide level.

Here, we report on the unexpected finding that the cS1 site is dispensable for K48 chain binding. Using a combination of hydrogen-deuterium exchange mass spectrometry (HDX-MS), NMR, chemical crosslinking, small-angle X-ray scattering (SAXS), and molecular docking, we find a cryptic K48 Ub chain-binding site located on the opposite face of UCH37 relative to the cS1 site. K48 linkage specificity is conferred by two regions: one composed of an aromatic-rich helix-loop-helix motif and another containing L181. Targeted mutagenesis reveals that the K48-specific sites are necessary for chain binding and removing K48 branchpoints, but not for C-terminal hydrolysis. By reconstituting proteasome complexes with mutants of UCH37 either defective in Ub binding at the cS1 site

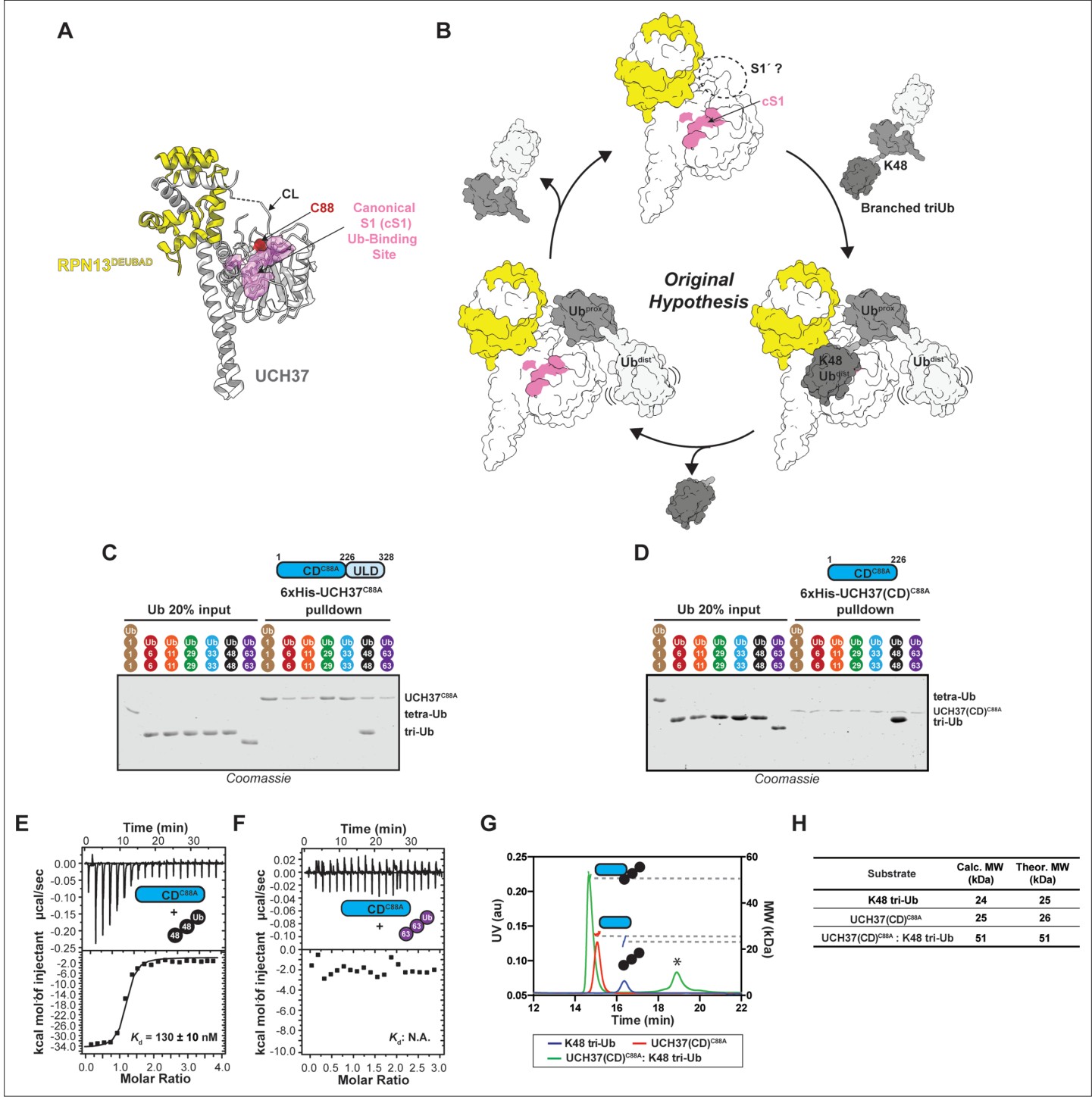

**Figure 1.** The catalytic domain (CD) of UCH37 confers K48 linkage specificity. (**A**) Structure of UCH37 bound RPN13[DEUBAD] (PDB ID: 4UEL; see also 4WLR) highlighting residues comprising the canonical S1 (cS1) Ubiquitin (Ub)-binding site (shown in pink). (**B**) Original hypothesis for how UCH37 uses the cS1 site along with an unknown S1' site to catalyze K48 chain debranching. (**B**) SDS-PAGE analysis of 6×His-UCH37[C88A]-mediated pulldowns with M1-, K6-, K11-, K29-, K33-, K48-, and K63-linked Ub chains. Each chain (50 pmol) was mixed with 6×His-UCH37[C88A] (5 nmol) immobilized on Ni-NTA resin. (**C**) SDS-PAGE analysis of 6×His-UCH37(CD)[C88A] with M1-, K6-, K11-, K29-, K33-, K48-, and K63-linked Ub chains. Each chain (50 pmol) was mixed with 6×His-UCH37(CD)[C88A] (5 nmol) immobilized on Ni-NTA resin. (**D–E**) Isothermal titration calorimetry analysis of UCH37(CD)[C88A] binding to K48 tri-Ub (**C**), and K63 tri-Ub. (**D**) (**F**) Size exclusion chromatography coupled with multiangle light scattering (SEC-MALS) analysis of the interaction between UCH37(CD)[C88A] and K48 tri-Ub. * denotes a UV peak without corresponding light scattering data. (**G**) Theoretical and calculated molar mass of complexes detected by SEC-MALS.

*Figure 1 continued on next page*

*Figure 1 continued*

The online version of this article includes the following source data and figure supplement(s) for figure 1:

**Source data 1.** ITC and SEC-MALs analysis of UCH37[C88A] and UCH37[C88A] (CD) with Ub chains.

**Figure supplement 1.** The catalytic domain (CD) of UCH37 confers K48 linkage specificity.

(frontside) or K48 chain binding (backside), we find that only the latter is required for regulating the degradation of substrates modified with branched chains. When the front- and backside mutants are expressed in cells lacking UCH37, the frontside mutant can rescue wild-type (WT) activity but the backside mutant cannot. With complete mutational control over the activity of UCH37, we are then able to identify branched chain-modified proteins whose abundance and turnover depend on K48 chain binding. Our results suggest that not all DUBs use a single S1 site to perform all the necessary biochemical functions.

## Results

### K48 linkage selectivity is embedded in the catalytic domain of UCH37

Previously, we found that UCH37 exclusively interacts with K48 chains regardless of overall architecture but only cleaves the K48 linkage if it is present at a branchpoint (*Deol et al., 2020*). UCH37 is composed of a catalytic domain (CD) and a C-terminal helical domain termed the UCH37-Like Domain (ULD). Whether both domains are required for K48 specificity is unclear. Thus, to address this issue, we first performed a series of pulldown experiments with a panel of homotypic Ub chains. Both inactive full-length UCH37 (UCH37[C88A]) and the inactive form of the CD (CD[C88A]) exclusively pulldown K48 chains (*Figure 1C-D*, *Figure 1—figure supplement 1A-C*). K48 linkage specificity is also observed with the CD by isothermal titration calorimetry (ITC) and binding affinity improves with increasing chain length (*Figure 1E-F*, *Figure 1—figure supplement 1D-E*). Size exclusion chromatography coupled with multiangle light scattering (SEC-MALS) shows the CD forms a 1:1 complex with a K48-linked Ub trimer (tri-Ub) (*Figure 1G-H*), indicating a single CD is necessary and sufficient for K48 specificity.

### HDX-MS reveals slower exchange on the backside of UCH37 upon binding K48 chains

To identify the K48 Ub chain-specific binding sites in UCH37, we turned to HDX-MS. We decided to use UCH37[C88A] bound to the DEUBAD (DEUBiquitinase ADaptor) domain of its partner protein RPN13 (referred to as the UCH37•RPN13[DEUBAD] complex). Like the CD, the UCH37•RPN13[DEUBAD] complex binds K48 chains with 1:1 stoichiometry (*Figure 1—figure supplement 1F-H*), but more importantly, the addition of RPN13[DEUBAD] releases UCH37 from an autoinhibited state thereby enhancing the binding affinity toward mono-Ub (*Hamazaki et al., 2006*; *Qiu et al., 2006*; *Sahtoe et al., 2015*; *Vander Linden et al., 2015*; *Yao et al., 2006*). This enables the investigation of a wide range of interactions from mono-Ub to branched trimers, which is necessary for uncovering cryptic Ub-binding sites outside of the cS1 site.

After obtaining initial data showing that high peptide coverage for UCH37 (97%) and RPN13[DEUBAD] (98%) can be achieved at a resolution of 7–14 amino acids, we moved on to a proof-of-concept experiment with mono-Ub. Specifically, we compared the deuterium uptake between mono-Ub bound UCH37[C88A]•RPN13[DEUBAD] and free UCH37[C88A]•RPN13[DEUBAD]. A heat map corresponding to the differential deuterium uptake plots of individual peptides of UCH37[C88A] shows a region near the C-terminus of the CD (residues 205–218) that undergoes slower exchange in the presence of Ub (*Figure 2A*). There is also slow exchange in the adjoining helical ULD domain (residues 257–270). The same regions are protected when Ub is covalently attached to the active site Cys (*Figure 2B*). Mapping these data on the existing structure (*Sahtoe et al., 2015*; *Vander Linden et al., 2015*) shows the protected regions are either adjacent to or directly encompass the cS1 site (*Figure 2E–F*).

We then shifted our attention to K48 chains. UCH37[C88A]•RPN13[DEUBAD] was mixed with either K48 di-Ub, tri-Ub, or K6/K48 tri-Ub and subjected to HDX-MS analysis (*Figure 2—figure supplement 1A–C*). Differential heat maps comparing the chain-bound to mono-Ub-bound form were calculated to identify sites undergoing slower exchange outside of the S1 site (*Figure 2C*, *Figure 2—figure*

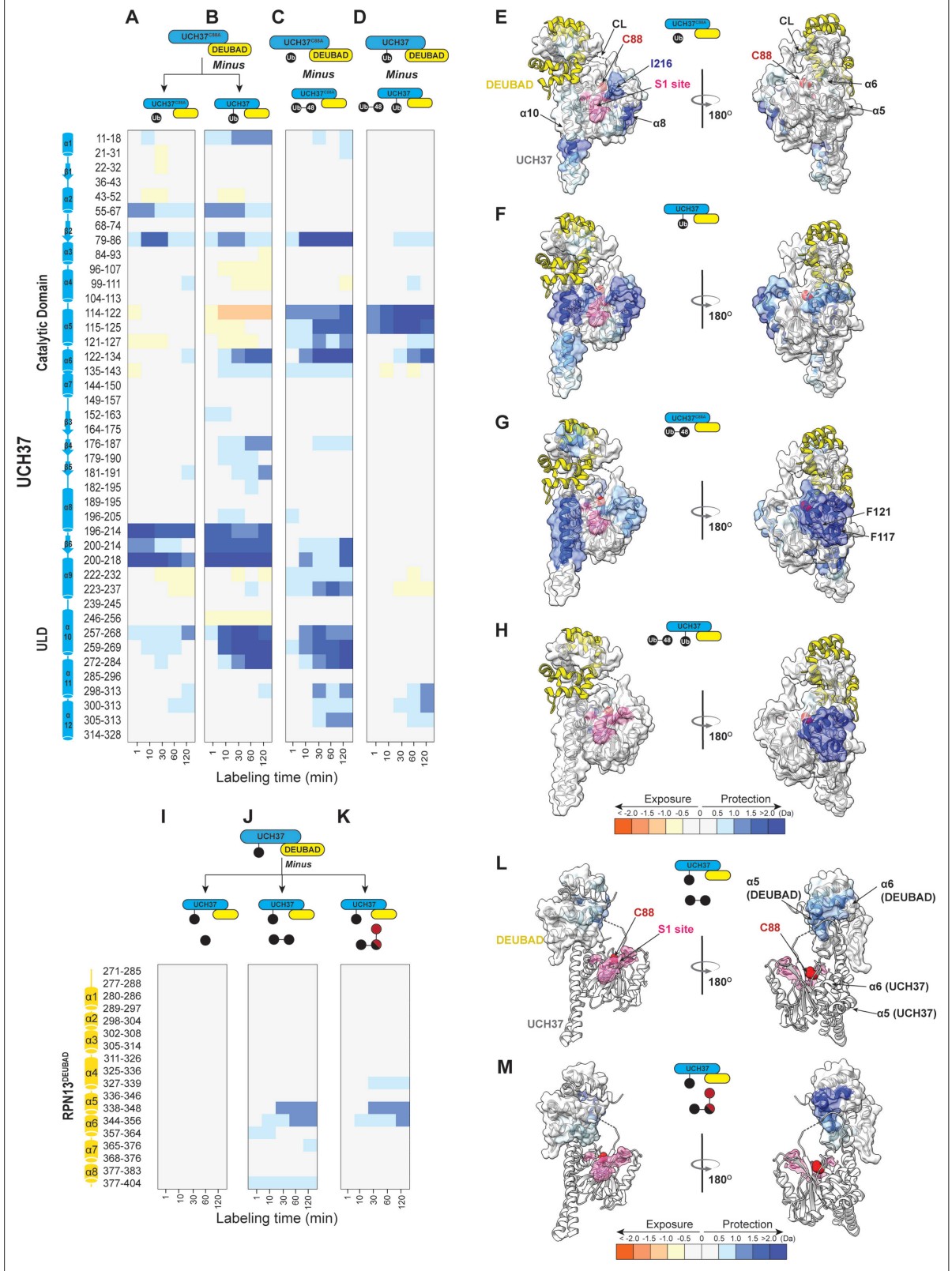

**Figure 2.** Hydrogen-deuterium exchange mass spectrometry (HDX-MS) uncovers a cryptic K48 chain-specific binding site. (**A**) Differential deuterium uptake plot comparing mono-ubiquitin (Ub) bound UCH37$^{C88A}$•RPN13$^{DEUBAD}$ to free UCH37$^{C88A}$ •RPN13$^{DEUBAD}$. (**B**) Differential deuterium uptake plot comparing UCH37•RPN13$^{DEUBAD}$ covalently linked to Ub propargylamine at the active site Cys (Ub~UCH37•RPN13$^{DEUBAD}$) to free UCH37$^{C88A}$ •RPN13$^{DEUBAD}$. (**C**) Differential deuterium uptake plot comparing K48 di-Ub-bound UCH37$^{C88A}$•RPN13$^{DEUBAD}$ to mono-Ub-bound UCH37$^{C88A}$•RPN13$^{DEUBAD}$. (**D**) Differential

*Figure 2 continued on next page*

*Figure 2 continued*

deuterium uptake plot comparing K48 di-Ub-bound Ub ~UCH37•RPN13[DEUBAD] to free Ub ~UCH37•RPN13[DEUBAD]. (**E–G**) Structure of UCH37 (PDB ID: 4UEL) showing regions with statistically significant differences in exchange upon noncovalent binding to mono-Ub (**E**), covalent attachment of Ub to the active site (**F**), and noncovalent binding to K48 di-Ub. (**G**) Data correspond to 2 hr of deuterium labeling. Other highlighted regions include the catalytic Cys (**C88**), the crossover loop (**CL**), and the canonical S1 site (pink). (**H**) Structure of UCH37 (PDB ID: 4UEL) showing statistically significant differences in exchange upon noncovalent binding of K48 di-Ub to Ub ~UCH37•RPN13[DEUBAD]. Data correspond to 2 hr of deuterium labeling. (**I–K**) Differential deuterium uptake plot comparing the effects of mono-Ub (**I**), K48 di-Ub (**J**), and K6/K48 tri-Ub (**K**) binding to Ub ~UCH37•RPN13[DEUBAD] on the exchange of residues in RPN13[DEUBAD]. (**L–M**) Heat map showing regions of RPN13[DEUBAD] with statistically significant differences in deuterium exchange upon noncovalent binding to K48 di-Ub (**L**), and noncovalent binding to K6/K48 tri-Ub (**M**). Data correspond to 2 hr of deuterium labeling.

The online version of this article includes the following source data and figure supplement(s) for figure 2:

**Source data 1.** HDX-MS analysis of UCH37 and RPN13DEUBAD.

**Figure supplement 1.** Hydrogen-deuterium exchange mass spectrometry uncovers a cryptic K48 chain-specific binding site.

**Figure supplement 2.** Isothermal titration calorimetry (ITC) and fluorescence polarization analysis of binding to ubiquitin (Ub) ~UCH37•RPN13[DEUBAD].

*supplement 1D–F*). From these data it is apparent that slow exchange in residues comprising helices α5 and α6 is exclusive to K48 chains (*Figure 2C*, *Figure 2—figure supplement 1D–F*). The α5–6 motif is located on the backside of UCH37 relative to the S1 site and is not predicted to interact with Ub (*Figure 2G*).

We envisioned two possible scenarios that could account for protection of the α5–6 motif. The first is that the distal K48-linked Ub subunit binds the S1 site while the proximal interacts with a portion of the CL. Examining the structures of mono-Ub bound to the S1 site of UCH37, there does not seem to be enough space between Ub and the CL to accommodate another Ub molecule. Thus, it is plausible that the C-terminal portion of the ULD acts as a hinge to allow for the DEUBAD domain and the CL to move closer to the α5–6 motif, decreasing its mobility and thus reducing the exchange rate. The second is that lower deuterium uptake could simply reflect the presence of a cryptic Ub-binding site.

If the α5–6 motif is the preferred binding site for K48 chains, then K48 chain binding should occur to a similar extent regardless of whether the S1 site is occupied because the α5–6 motif is on a distinct face. However, if S1 together with the CL forms a K48 specific binding motif, then covalent attachment of Ub to the active site Cys should block binding to K48 chains. Surprisingly, we find that mono-Ub still binds Ub ~UCH37•RPN13[DEUBAD], albeit with twofold lower affinity compared to the apo complex, and the binding of K48 di-Ub is not altered at all (*Figure 2—figure supplement 2A–B*). HDX data support these results. Weak but reproducible protection of the α5–6 motif is observed upon addition of mono-Ub to Ub ~UCH37•RPN13[DEUBAD], and the degree of protection becomes much more pronounced when either K48 di-Ub or K6/K48 tri-Ub is added (*Figure 2D*, *Figure 2—figure supplement 1G–H*). It appears that when Ub is conjugated to UCH37, binding of either K48 di-Ub or K6/K48 tri-Ub affords stronger protection of the α5–6 motif (*Figure 2D*, *Figure 2—figure supplement 1H*). This could be due to an allosteric effect resulting in enhanced binding or a situation in which K48 chains only interact with the backside because sampling of both faces is shutdown. Binding data support the latter by showing that Ub conjugation does not improve the affinity toward K48 di-Ub (*Figure 2—figure supplement 2C–D*) or K6/K48 tri-Ub (*Figure 2—figure supplement 2E–F*). Together, these data indicate the cS1 site is dispensable for binding K48 chains and suggest that specificity is conferred through interactions mediated by the α5–6 motif on the backside of UCH37.

The reciprocal HDX data on the DEUBAD domain is also informative. Titrating in K48 di-Ub or K6/K48 tri-Ub to Ub~UCH37[C88A]•RPN13[DEUBAD] results in slower exchange in two partially overlapping peptides—338–348 and 344–356—corresponding to helices 5 and 6 of the DEUBAD domain (*Figure 2J–K*). The exchange rate for the same set of residues is not perturbed by mono-Ub (*Figure 2I*), indicating that slower deuterium uptake is unique to K48 chains. As the existing structure of Ub~UCH37•RPN13[DEUBAD] shows, α5–6 of the DEUBAD interacts with the CL to form a contiguous surface facing the backside of the enzyme (*Figure 2L–M*). Thus, the DEUBAD along with the CL could provide one Ub-binding site while the α5–6 motif of UCH37 provides another. Alternatively, the DEUBAD could restrict the motion of the CL to afford a conformation conducive to chain binding.

Although UCH37 binds K48 chains independent of chain architecture, branched chains are cleaved with higher efficiency than homotypic K48 chains (*Deol et al., 2020*; *Shorkey et al., 2021*; *Song et al., 2021*). Thus, we expected to find distinct regions that become more protected in the presence

of branched chains. Instead, we only observe differences in the degree of deuterium uptake, as the exchange rates within the same regions of protection are faster with K6/K48 tri-Ub relative to K48 di- and tri-Ub (*Figure 2—figure supplement 2G*). These results suggest the interaction between UCH37•RPN13^DEUBAD and branched chains is more dynamic than that with unbranched K48 chains.

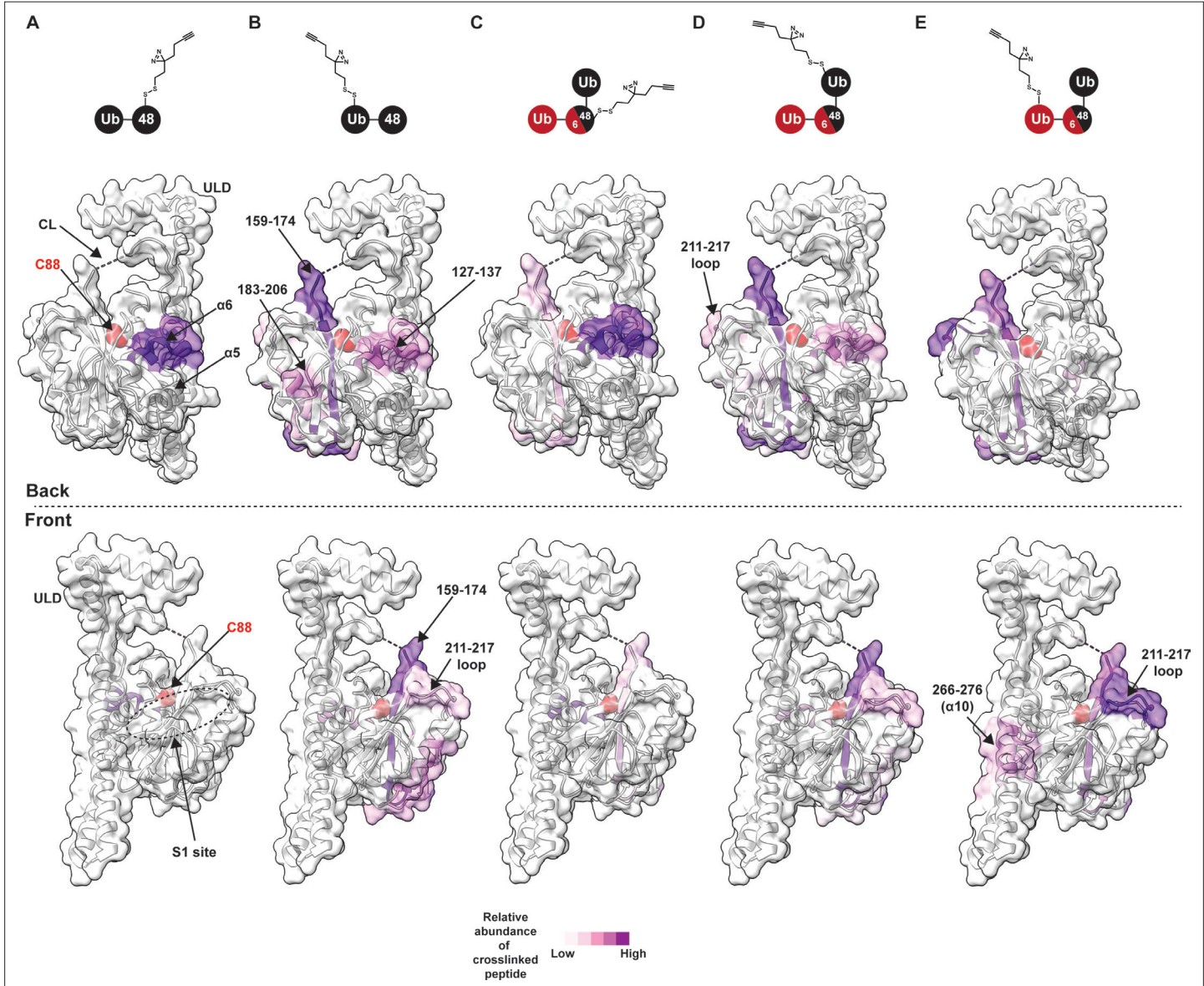

**Figure 3.** Chemical crosslinking confirms the presence of a K48 chain-specific binding site on the backside of UCH37. (**A–E**) A photolabile diazirine-based crosslinker was appended to individual ubiquitin (Ub) subunits of K48 chains. Crosslinked peptides were mapped onto the structure of UCH37 (PDB: 4UEL) according to their relative abundance based on the area under the curve of the extracted ion chromatogram. (**A**) Map of the crosslinking data for K48 di-Ub binding to UCH37^C88A•RPN13^DEUBAD with the proximal Ub subunit labeled with the diazirine colored by normalized relative abundance of crosslinked peptide. (**B**) Map of the crosslinking data for K48 di-Ub binding to UCH37^C88A•RPN13^DEUBAD with the distal Ub subunit labeled with the diazirine colored by normalized relative abundance of crosslinked peptide. (**C**) Map of the crosslinking data for K6/K48 tri-Ub binding to UCH37^C88A•RPN13^DEUBAD with the proximal Ub subunit labeled with the diazirine colored by normalized relative abundance of crosslinked peptide. (**D**) Map of the crosslinking data for K6/K48 tri-Ub binding to UCH37^C88A•RPN13^DEUBAD with the K48-linked distal Ub subunit labeled with the diazirine colored by normalized relative abundance of crosslinked peptide. (**E**) Map of the crosslinking data for K6/K48 tri-Ub binding to UCH37^C88A•RPN13^DEUBAD with the K6-linked distal Ub subunit labeled with the diazirine colored by normalized relative abundance of crosslinked peptide.

The online version of this article includes the following figure supplement(s) for figure 3:

**Figure supplement 1.** Chemical crosslinking confirms the presence of a K48 chain-specific binding site on the backside of UCH37.

## Backside binding is confirmed by chemical crosslinking

If the backside affords a K48-specific binding surface, we hypothesized that individual Ub subunits would crosslink distinct sites. To test this, we installed a chemical crosslinker on individual subunits of K48 di-Ub (*Figure 3A–B*) and K6/K48 tri-Ub (*Figure 3C–E*). We decided to place a photolabile, diazirine moiety (*Lin et al., 2021*) at position-46, as this is close to the I44 hydrophobic patch, which forms a hotspot for many interactions (*Husnjak and Dikic, 2012*). For comparison, mono-Ub was also modified with the diazirine crosslinker at the same position (*Figure 3—figure supplement 1A*). As expected, irradiation results in the formation of crosslinked products with each Ub variant (*Figure 3— figure supplement 1B*).

After band extraction and subsequent digestion, crosslinked sites on UCH37 were identified by MS (*Figure 3—figure supplement 1C*). The relative abundance of each site was estimated from the area under the curve of the extracted ion chromatogram. Mapping these data onto the structure of UCH37 reveals that α10 of the ULD is the primary crosslinking site when the diazirine is at position-46 of mono-Ub (*Figure 3—figure supplement 1D*). This result is entirely consistent with the Ub-bound structure, which shows A46 of Ub pointing directly toward α10. Peptides corresponding to α6, the C-terminal end of the CL, and 211–217 loop are also trapped by the diazirine-modified mono-Ub, but to a much lesser extent compared to the α10 peptide. These results confirm that mono-Ub also binds the backside α5–6 motif in addition to the cS1 site.

In accord with the HDX-MS data, K48 chains shift the preferred crosslinking sites to the backside of UCH37. The proximal subunit of K48 di-Ub (Ub$^{prox}$) almost exclusively crosslinks a peptide corresponding to α6 (*Figure 3A*). The distal subunit (Ub$^{dist}$), which generally displays weaker crosslinking compared to the proximal Ub, traps several sites (*Figure 3B* and *Figure 3—figure supplement 1C*). The most prevalent is near the C-terminal end of the CL. Other less abundant sites include α6, and a peptide fragment comprising residues 183–206. Crosslinked peptides corresponding to RPN13$^{DEUBAD}$ were not detected. Our crosslinking data together with HDX-MS therefore uncover two distinct regions on the backside of UCH37 that are involved in the interaction with unbranched K48 chains: the α5–6 motif and a more diffuse region composed of the C-terminal end of the CL along with the 176–190 and 211–217 loops. These data also suggest there is preferred orientation for the two subunits of K48 di-Ub, with Ub$^{prox}$ binding α5–6 and Ub$^{dist}$ interacting with a series of flexible loops near the C-terminus of the CL.

We then wanted to know whether the K48-linked portion of a branched chain adopts a similar orientation. Like the Ub$^{prox}$ from K48 di-Ub, we found that the diazirine at position-46 of the Ub$^{prox}$ from K6/K48 tri-Ub shows a strong preference for crosslinking the α6 region of UCH37 (*Figure 3C*), but there is significantly more crosslinking at the C-terminus of CL compared to the unbranched chain. The crosslinking pattern for the K48-linked Ub$^{dist}$ also resembles the Ub$^{dist}$ of the dimer (*Figure 3D*); however, there are subtle differences in the degree to which peptides from 183 to 206 and 211–217 are crosslinked (*Figure 3B and D*, and *Figure 3—figure supplement 1C*). Unbranched and branched chains might therefore engage UCH37 in a similar manner, but the K48-linked portion of a branchpoint might not be confined to an orientation in which the proximal subunit is localized to the α5–6 motif. As a result, the K48-linked subunits of branched chain could be more prone to swapping positions.

The non-K48-linked subunit at a branchpoint could further enable subunits to swap positions. The crosslinking data shows that the 211–217 loop is one of the most prevalent sites trapped by the K6-linked Ub$^{dist}$ of K6/K48 tri-Ub (*Figure 3E*). For position-46 of the K6-linked Ub$^{dist}$ to be projected toward residues 211–217, Ub$^{prox}$ would have to be positioned near the C-terminus of the CL thereby placing the K48-linked Ub$^{dist}$ at the α5–6 motif. If K48 cleavage occurs with a branched chain in this configuration, the α5–6 motif would ostensibly be the S1 site dedicated to isopeptidase activity. Regardless of the precise orientation of the two K48-linked subunits, our data unexpectedly place the scissile bond on the opposite face relative to the cS1 site (*Figure 4H*), suggesting cleavage occurs from the backside.

## Structural models of K48 di-Ub bound to UCH37•RPN13$^{DEUBAD}$

Building on the results from HDX-MS and crosslinking, we sought to generate structural models of K48 chains bound to UCH37•RPN13$^{DEUBAD}$. Molecular dynamics (MD) simulations were performed to sample the conformational space of UCH37•RPN13$^{DEUBAD}$ (*Figure 4—figure supplement 1A*). A total of three conformations were selected based on fitting to experimental SAXS data

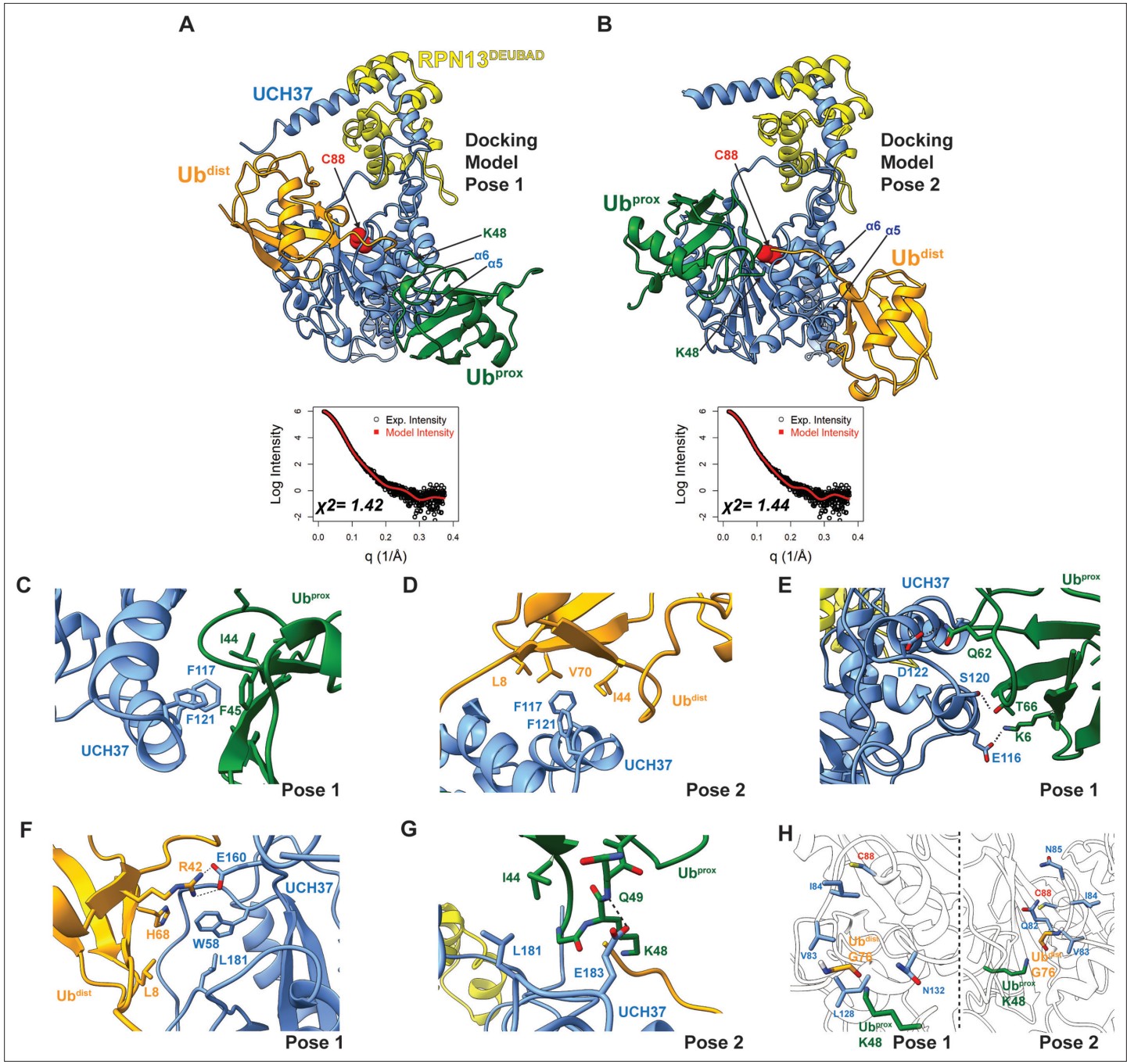

**Figure 4.** Docking models of the K48 di-ubiquitin (Ub):UCH37•RPN13[DEUBAD] complex. (**A–B**) HADDOCK docking models show two poses corresponding to the interaction between K48 di-Ub and UCH37 along with their fit to experimental small-angle X-ray scattering data of the K48 di-Ub:UCH37[C88A]•RPN13[DEUBAD] complex. The goodness of fit to the experimental intensity is represented by $\chi^2$ values. In the first pose (**A**), the proximal Ub (Ub[prox]; green) interacts with α5–6 of UCH37. In the second pose (**B**), the distal Ub (Ub[dist]; orange) interacts with α5–6 of UCH37. (**C**) Residues highlighting the interaction between the aromatic-rich region of UCH37 α5–6 and the I44 patch of Ub[prox] in pose 1 (**A**). (**D**) Residues highlighting the interaction between the aromatic-rich region of UCH37 α5–6 and the I44 patch of Ub[dist] in pose 2 (**B**). (**E**) Polar contacts between Ub[prox] and UCH37 α5–6 in pose 1 (**A**). (**F**) Contacts between Ub[dist] and residues of UCH37 located outside the α5–6 motif in pose 1 (**A**). (**G**) Contacts between Ub[prox] and residues of UCH37 located outside the α5–6 motif in pose 2 (**B**). (**H**) The relative location of active site and the scissile, K48 isopeptide bond in molecular docking poses 1 and 2. In pose 1, residues of α6 and the loop leading into the catalytic Cys (C88) form a barrier for the isopeptide bond. In pose 2, only Q82 of UCH37 blocks the K48 isopeptide bond.

The online version of this article includes the following figure supplement(s) for figure 4:

**Figure supplement 1.** Structural models of the K48 di-ubiquitin (Ub):UCH37•RPN13[DEUBAD] complex.

*Figure 4 continued on next page*

eLife Research article

Biochemistry and Chemical Biology

**Figure supplement 2.** Small-angle X-ray scattering (SAXS) of the K48 di-ubiquitin (Ub):UCH37•RPN13[DEUBAD] complex.

**Figure supplement 3.** Contact frequency analysis of the K48 di-ubiquitin (Ub):UCH37•RPN13[DEUBAD] complex.

(*Figure 4—figure supplement 1B–C*). Each conformer then served as a starting point for docking K48 di-Ub using the program HADDOCK (*Honorato et al., 2021*; *van Zundert et al., 2016*). A complete list of docking restraints can be found in *Table 1*. The top-scoring solutions were further evaluated based on their agreement with the SAXS profile of K48 di-Ub:UCH37[C88A]•RPN13[DEUBAD] (*Figure 4—figure supplement 1D*, *Figure 4—figure supplement 2A–D*). Excellent fits were obtained for three different poses (*Figure 4A–B* and *Figure 4—figure supplement 2E*). In the first two, one subunit of K48 di-Ub forms an interface with the α5–6 motif, and the other subunit is positioned near the C-terminal end of the CL. The K48 dimer adopts an extended conformation in each of the docking models. In the third pose, one of the Ub subunits can be seen interacting with the CL and the DEUBAD domain, which is consistent with the HDX data (*Figure 2J–K*). Overall, the bulk of the structures corresponds to molecular docking poses 1 and 2, while pose 3 represents a minor fraction.

Taking all the best-fitting docking poses into account (59 in total), we wanted to obtain more insight into the specific sites involved in the interaction with K48 chains. We therefore measured the frequency with which residues of UCH37 and Ub come into close contact (*Figure 4—figure supplement 3A–B*). Residues deemed significant are two standard deviations from the mean. Based on this analysis, it is evident that the conserved aromatic residues F117 and F121 form the nexus of the α5–6 motif (*Figure 4—figure supplement 3A*). This aromatic-rich region interacts with the I44 patch of Ub regardless of the orientation of K48 di-Ub (*Figure 4—figure supplement 3B*). When Ub[prox] is docked at α5–6 (pose 1), I44 and F45 form contacts with both F117 and F121 of UCH37 (*Figure 4C*). Swapping Ub[prox] with Ub[dist] (pose 2), forces L8, I44, and V70 of Ub to interact with F117 and F121 (*Figure 4D*). The preference for Ub[prox] binding to α5–6 could, in part, stem from polar contacts, as there are three residues in α5–6 (E116, S120, and D122) predicted to engage Ub[prox] and only two for Ub[dist] (*Figure 4E*).

L181 of UCH37 appears to play an integral role in the second Ub-binding site. In one docking model, L181 contacts L8 of Ub[dist], and in another it is near I44 of Ub[prox] (*Figure 4F–G*). Additional contacts are established with other residues located in different loop regions, for example, E160 and W58 (*Figure 4F*), or with a backbone amide of Ub[prox] (*Figure 4G*). These models suggest that binding to both backside sites can only be accomplished with the K48 linkage, as other linkages do not place the two I44 patches in the correct orientation for engagement. The models also suggest cleavage of the K48 isopeptide bond could occur from the backside if there is movement of residues surrounding the active site (e.g. Q82 in pose 2; *Figure 4H*).

**Table 1.** Active and passive restraints used for HADDOCK modeling.

| | UCH37 | DEUBAD |
|---|---|---|
| Active residues | 117, 121, 154 | |
| Passive residues | 81, 82, 83, 84, 85, 118, 119, 120, 122, 123, 124, 125, 126, 127, 128, 155, 156, 157, 158, 159, 160, 161, 162, 181, 182, 183, 184 | 403, 406, 410, 419, 422 |
| Flexible residues | | 329–347, 444–468 |
| | Ub2K48-proximal subunit | Ub2K48-proximal subunit |
| Active residues | 8, 44, 70 | 8, 44 |
| Passive residues | 36, 42, 43, 45, 46, 49, 50, 57, 58, 59, 60, 68, 71, 72, 73, 74, 75, 76 | 42, 43, 45, 46, 47, 48, 49, 50, 57, 58, 59, 60, 68, 70, 71 |
| Flexible residues | 48 | 72–76 |

## An aromatic-rich motif in α5–6 is required for backside binding

Targeted mutagenesis confirms the importance of the α5–6 motif and L181 in K48-specific binding. Replacing two of the key residues of the α5–6 motif, F117 and F121, with alanine (F117A and F121A) reduces the binding affinity toward K48 di-Ub to mono-Ub levels (*Figure 5—figure supplement 1A–B*) while retaining the same Ub-7-Amino-4-methylcoumarin (AMC) hydrolytic activity as WT UCH37 (*Figure 5—figure supplement 1C*). The only other Ala substitution with a similar effect as F117A and F121A is L181 (*Figure 5—figure supplement 1C–E*). According to conservation analysis, F117 and L181 are invariably present in UCH37 homologs (*Figure 5—figure supplement 1F*).

HDX-MS and NMR data further underscore the importance of F117 in backside binding. HDX-MS analyses of UCH37$^{C88A/F117A}$•RPN13$^{DEUBAD}$ (the F117A complex) show that the region encompassing residues 115–125 is no longer protected from exchange when either mono-Ub or K48 di-Ub is added (*Figure 5A*). Moreover, in contrast to UCH37$^{C88A}$•RPN13$^{DEUBAD}$ (the C88A complex; *Figure 5B–C*), the F117A complex induces few perturbations in the Ile-region of the $^{1}$H$^{13}$C-methyl TROSY spectra of Isoleucine/Leucine/Valine (ILV)-labeled K48-linked Ub$^{prox}$ and Ub$^{dist}$ (*Figure 5D–E*), indicating that the dimer largely remains in the unbound state. Binding to the frontside, however, is retained by the F117A complex, as evidenced by the nearly indistinguishable bound-state peaks for $^{1}$H$^{13}$C-methyl, ILV-labeled mono-Ub with the C88A and F117A complexes (*Figure 5F*). There are also minor peaks in the spectra of Ub$^{prox}$ and Ub$^{dist}$ with the F117A complex resembling the resonances for the bound state of mono-Ub (*Figure 5D–E*), suggesting that when the backside is inaccessible, each subunit can interact with the frontside to some extent.

One of the challenges in assessing the contribution of individual Ub residues to the interaction with UCH37 using NMR is that front- and backside binding are conflated. This issue is certainly germane to Ub$^{prox}$ considering the spectrum of this subunit in the presence of the C88A complex is nearly identical that of mono-Ub bound to the same complex. The spectra collected for Ub$^{dist}$, however, tell a different story. A marked upfield shift in the $^{1}$H dimension is observed for Ub$^{dist}$ I44 in the presence of the C88A complex (*Figure 5C*). The same peak is completely absent from the spectrum of Ub$^{dist}$ bound to the F117A complex (*Figure 5E*). These data suggest the shift observed for Ub$^{dist}$ I44 is a signature of backside binding and that I44 of the distal subunit plays an important role in the backside interaction.

## Loss of the aromatic-rich patch impairs chain debranching and proteasomal degradation

Our data thus far demonstrate the importance of the α5–6 motif in binding, but do not address its role in the catalytic activity of UCH37. To this end, we monitored the cleavage of a branched trimer substrate (K6/K48 tri-Ub) over time by following the formation of di- and mono-Ub using SDS-PAGE. The steady-state parameters reveal that F117A and F121A each reduce the debranching activity by ~20- fold; L181A has a similar effect (*Figure 6A*). F117A and F121A also compromise the cleavage of high-molecular weight (HMW) Ub chains bearing different branchpoints (*Figure 6B*). These results indicate that impaired binding to K48 chains translates to a marked decrease in debranching activity.

We then asked whether a defective S1 site affects chain debranching. By engaging I36 of Ub, I216 plays an integral role in the S1 site. Replacing I216 with Glu (I216E) diminishes the hydrolytic activity toward Ub-AMC by ~11-fold, consistent with previous reports (*Figure 5—figure supplement 1C*; *Sahtoe et al., 2015*). Although the C88A mutant enhances binding to mono-Ub and Ub chains (*Morrow et al., 2018*), the addition of I216E does not affect mono-Ub nor K48 di-Ub binding (*Figure 6—figure supplement 1A*). Chain debranching is also unaffected by the I216E substitution (*Figure 6A–B*), suggesting either the cS1 site is not involved in the isopeptidase activity or a Ub moiety is situated differently than what is predicted by the existing mono-Ub-bound structures.

To test whether the backside is necessary for the role of UCH37 on the proteasome, 26 S proteasomes deficient of WT UCH37 and RPN13 (ΔUCH37/ΔRPN13 Ptsms) were reconstituted with different RPN13-bound forms of UCH37: WT, inactive C88A, F117A, F121A, or I216E (*Figure 6—figure supplement 1B–J*). In the presence of K6/K48 and K48/K63 HMW Ub chains, WT- and I216E-replenished (rWT and rI216E) Ptsms generate smaller Ub conjugates, but the results with F117A- and F121A-replenished (rF117A and rF121A) Ptsms essentially mirror those with the inactive rC88A variant (*Figure 6C*). Degradation is also affected by F117A and F121A. Using K11/K48-polyUb-UBE2S-UBD and K48/K63-polyUb-titin-I27$^{V15P}$ as substrates, a decrease in polyubiquitinated species and a concomitant increase in Ptsm-derived peptides are observed with rWT and rI216E Ptsms but not with rF117A,

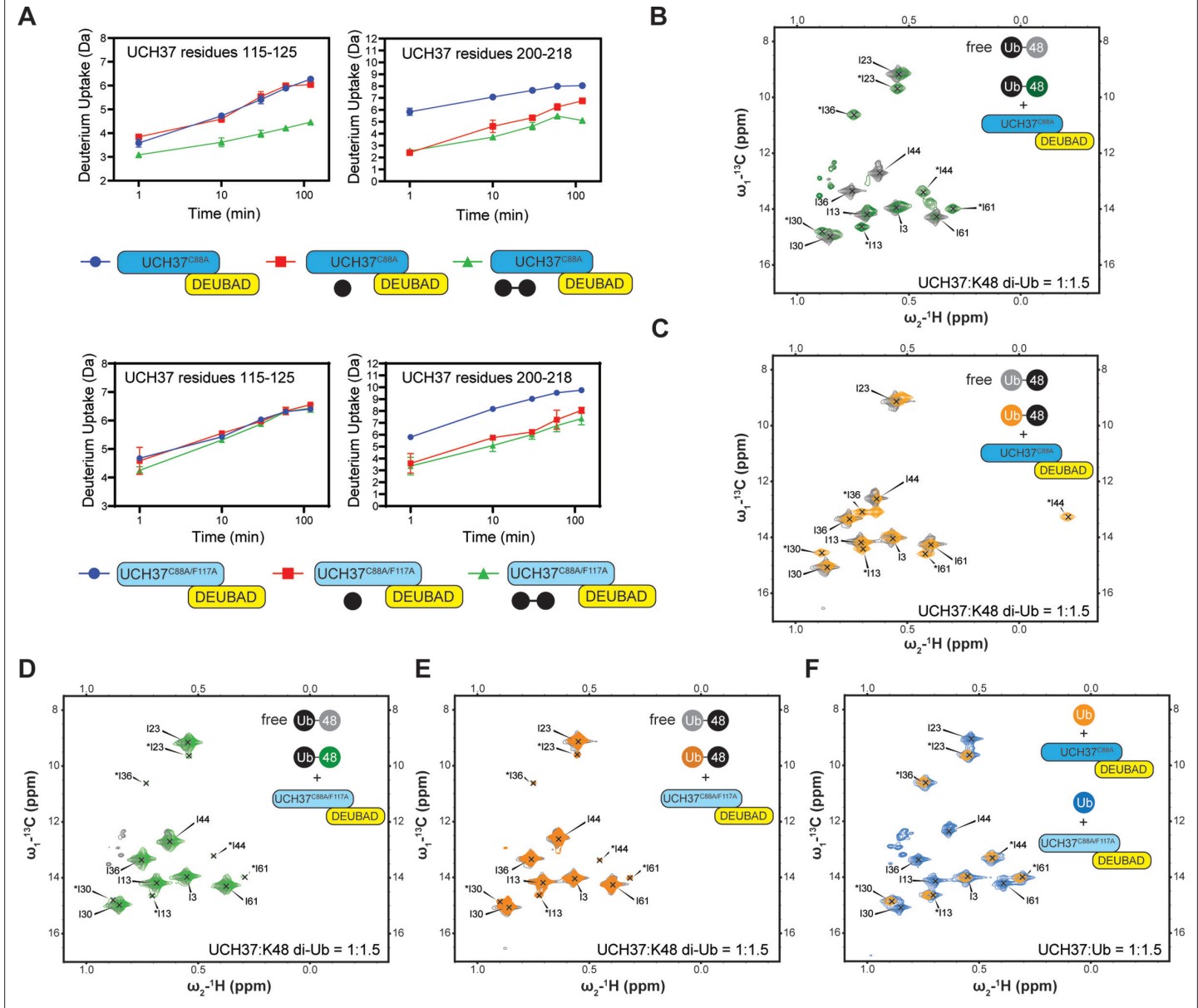

**Figure 5.** Backside mutant impairs K48 chain binding. (**A**) Deuterium uptake plots showing how the F117A mutation affects hydrogen-deuterium exchange in a peptide corresponding to residues 115–125 located in the α5–6 motif of UCH37. For comparison, the uptake plots corresponding to a region located outside the α5–6 motif (residues 200–218) are also shown. The data on the top represents the rate of exchange with UCH37$^{C88A}$•RPN13$^{DEUBAD}$ and the data on the bottom corresponds to UCH37$^{C88A/F117A}$•RPN13$^{DEUBAD}$. At least two replicates of each experiment were performed. (**B**)$^1$H$^{13}$C-methyl TROSY NMR spectra of the Ile region of ILV-labeled K48 di-ubiquitin (Ub) proximal subunit (Ub$^{prox}$) free in solution (gray) and bound to UCH37$^{C88A}$•RPN13$^{DEUBAD}$ (the C88A complex; green). Ratio of UCH37 to K48 di-Ub is 1:1.5. Concentrations used: 45 µM UCH37$^{C88A}$•RPN13$^{DEUBAD}$ and 30 µM K48 di-Ub. (**C**)$^1$H$^{13}$C-methyl TROSY NMR spectra of the Ile region of ILV-labeled K48 di-Ub distal subunit (Ub$^{dist}$) free in solution (gray) and bound to the C88A complex (orange). Ratio of UCH37 to K48 di-Ub is 1:1.5. Concentrations used: 45 µM UCH37$^{C88A}$ •RPN13$^{DEUBAD}$ and 30 µM K48 di-Ub. (**D**) $^1$H$^{13}$C-methyl TROSY NMR spectra of the Ile region of K48-linked Ub$^{prox}$ free in solution (gray) and bound to UCH37$^{C88A/F117A}$•RPN13$^{DEUBAD}$ (the F117A complex; green). Ratio of UCH37 to K48 di-Ub is 1:1.5. Concentrations used: 45 µM UCH37$^{C88A/F117A}$•RPN13$^{DEUBAD}$ and 30 µM K48 di-Ub. (**E**)$^1$H$^{13}$C-methyl TROSY NMR spectra of the Ile region of K48-linked Ub$^{dist}$ free in solution (gray) and bound to the F117A complex (orange). Ratio of UCH37 to K48 di-Ub is 1:1.5. Concentrations used: 45 µM UCH37$^{C88A/F117A}$•RPN13$^{DEUBAD}$ and 30 µM K48 di-Ub. (**F**)$^1$H$^{13}$C-methyl TROSY NMR spectra of the Ile region of mono-Ub bound to the C88A complex (orange) and the F117A complex (blue). Ratio of UCH37 to mono-Ub is 1:1.5. Concentrations used: 45 µM UCH37 •RPN13$^{DEUBAD}$ complex and 30 µM mono-Ub.

The online version of this article includes the following source data and figure supplement(s) for figure 5:

**Source data 1.** Full NMR spectra of mono-Ub and K48 di-Ub in presence and absence of UCH37.

**Figure supplement 1.** Backside mutant impairs K48 chain binding.

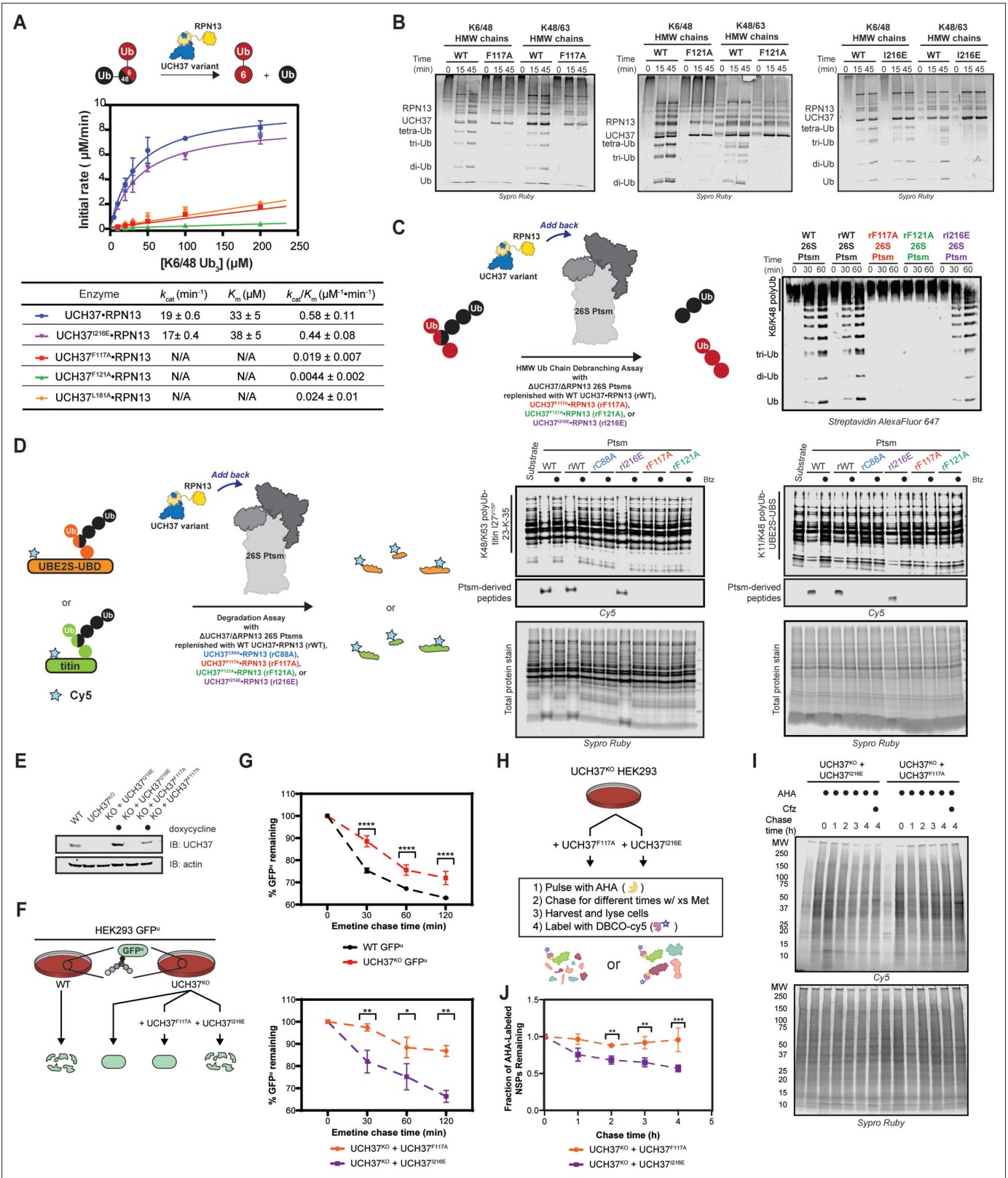

**Figure 6.** Backside mutants impair K48 chain debranching and UCH37-dependent proteasomal degradation. (**A**) Steady-state kinetic analyses of the cleavage of K6/K48 tri-ubiquitin (Ub) with different UCH37 variants (WT, F117A, F121A, and I216E) in complex with RPN13. The data corresponding to WT and I216E were fit to the nonlinear Michaelis-Menten equation. The linear equation, rate = $k_{cat}/K_m$[K6/K48 tri-Ub][Enzyme], was sufficient to fit the data corresponding to L181A, F117A, and F121A. (**B**) SDS-PAGE analysis of the cleavage of K6/K48 and K48/K63 high-molecular weight (HMW) Ub

*Figure 6 continued on next page*

*Figure 6 continued*

chains with different UCH37 variants (WT, F117A, F121A, and I216E) in complex with RPN13. Gels were imaged by Sypro Ruby staining. (**C**) Western blot analysis of cleavage of K6/K48 HMW Ub chains by WT Ptsm, WT UCH37•RPN13-replenished Ptsm (rWT), UCH37$^{F117A}$•RPN13-replenished Ptsms (rF117A), UCH37$^{F121A}$•RPN13-replenished Ptsms (rF121A), and UCH37$^{I216E}$•RPN13-replenished Ptsms (rI216E). 0.2 mg of Ptsm complex was used in each reaction. Immunoblotting was performed with Alexa Fluor 647-conjugated streptavidin. (**D**) Fluorescent SDS-PAGE analyses of the degradation of Cy5-labeled, K48/K63-Ub$_n$-titin-I27$^{V15P}$-23-K-35 and K11/K48-polyUb-UBE2S-UBD using WT, rWT, rC88A, rI216E, rF117A, and rF121A Ptsm complexes. Reactions were performed under multiturnover conditions and quenched after 1 hr. Gels stained for total protein are shown for reference. (**E**) Western blot analysis of UCH37 levels in WT, UCH37$^{KO}$ (KO), and KO HEK293 cells expressing UCH37 variants under a doxycycline-inducible promoter. Immunoblotting was performed with α-UCH37 and α-actin antibodies. (**F**) Schematic showing the different degradation profiles of the Ptsm activity reporter GFP$^u$ in WT and UCH37$^{KO}$ HEK293 cells expressing UCH37 variants under a doxycycline-inducible promoter. (**G**) Plots showing percent GFP$^u$ remaining in the indicated cell lines after shutting off translation with emetine. (**H**) Scheme showing the labeling and visualization of newly synthesized proteins (NSPs) with azidohomoalanine (AHA) in UCH37$^{KO}$ cells expressing either UCH37$^{I216E}$ or UCH37$^{F117A}$. (**I**) Fluorescent SDS-PAGE analysis of the turnover of AHA-labeled NSPs in UCH37$^{KO}$ cells expressing either UCH37$^{I216E}$ or UCH37$^{F117A}$. (**J**) Quantitation of the results shown in (**I**).

The online version of this article includes the following source data and figure supplement(s) for figure 6:

**Source data 1.** Emetine chase analysis and AHA pulse chase analysis of WT, UCH37$^{KO}$ and UCH37 mutants (F117A or I216E) transduced cells.

**Source data 2.** Uncropped gel images of gel-based branched tri-Ub kinetic assay, HMW Ub chain cleavage analysis, Western blot analysis of in vitro Ptsm degradation assay, and fluorescent gel analysis of AHA pulse chase assay.

**Figure supplement 1.** Backside mutants impair K48 chain debranching and UCH37-dependent proteasomal degradation.

rF121A, and rC88A Ptsms (*Figure 6D*). Because the rC88A Ptsms retain the ability to bind K48 chains, these results imply that the loss of chain debranching, not just chain binding, leads to the failure to degrade branched polyubiquitinated proteins.

The in vitro results suggest that degradation of branched chain-modified proteins depends on the K48-specific binding site of UCH37, raising the question of whether degradation in cells also depends on the same backside motif. Since the results with UCH37$^{F121A}$ mirror those with UCH37$^{F117A}$, we focused on the latter in our cellular assays. Either UCH37$^{I216E}$ or UCH37$^{F117A}$ was ectopically expressed from a doxycycline-inducible promoter in UCH37$^{KO}$ cells stably expressing the Ptsm activity reporter GFP$^u$ (*Figure 6E*). Translation shutoff experiments were then performed to measure the turnover of GFP$^u$. For comparison, the same experiments were conducted with WT HEK293 cells and UCH37$^{KO}$ cells as previously reported. In accord with the biochemical assays, UCH37$^{I216E}$ restores GFP$^u$ turnover in the null background, whereas the expression of UCH37$^{F117A}$ is unable to rescue the WT phenotype (*Figure 6F*). Of note, the degradation of GFP$^u$ is lower in cells expressing UCH37$^{F117A}$ compared to the null background. This could be due to an inhibitory effect caused by the inability to efficiently clear Ptsm-bound Ub chains that are removed from the substrate during translocation into the proteolytic chamber (see Discussion).

In an orthogonal assay, we examined the effect of UCH37$^{I216E}$ and UCH37$^{F117A}$ on the turnover of newly synthesized proteins (NSPs). Azidohomoalanine (AHA) is incorporated into NSPs in lieu of methionine during a short pulse to allow for visualization (*Figure 6G*; *Dieterich et al., 2006*; *Howden et al., 2013*; *McShane et al., 2016*). Degradation is then monitored after replenishing the growth medium with excess methionine. In agreement with the GFP$^u$ data, we found that AHA-labeled NSPs dissipate over time when UCH37$^{KO}$ cells are complemented with UCH37$^{I216E}$ but not with UCH37$^{F117A}$ (*Figure 6G*).

## Uncovering the targets of UCH37-dependent chain editing

Having complete mutational control over UCH37 activity in a null background allows us to identify proteins whose turnover is specifically affected by the loss of K48 chain-specific binding and debranching. To this end, we started by measuring total protein levels in unperturbed WT and UCH37$^{KO}$ cells; however, the differences between the two cell lines turned out to be minimal (*Figure 7—figure supplement 1A–B*). We then decided to treat cells with $H_2O_2$, as UCH37 has previously been implicated in the cellular response to oxidative stress (*Harris et al., 2019*).

In the presence of $H_2O_2$, two separate tandem mass tagging (TMT)-based proteomic experiments were performed: one comparing the proteomes of UCH37$^{F117A}$- and UCH37$^{WT}$-expressing UCH37$^{KO}$ cells, and another comparing the proteomes of UCH37$^{I216E}$- and UCH37$^{WT}$-expressing UCH37$^{KO}$ cells (*Figure 7A*). This setup allowed us to analyze duplicate or triplicate samples of each $H_2O_2$-treated cell line in a single experiment and eliminate strain background effects. High-resolution MS quantified

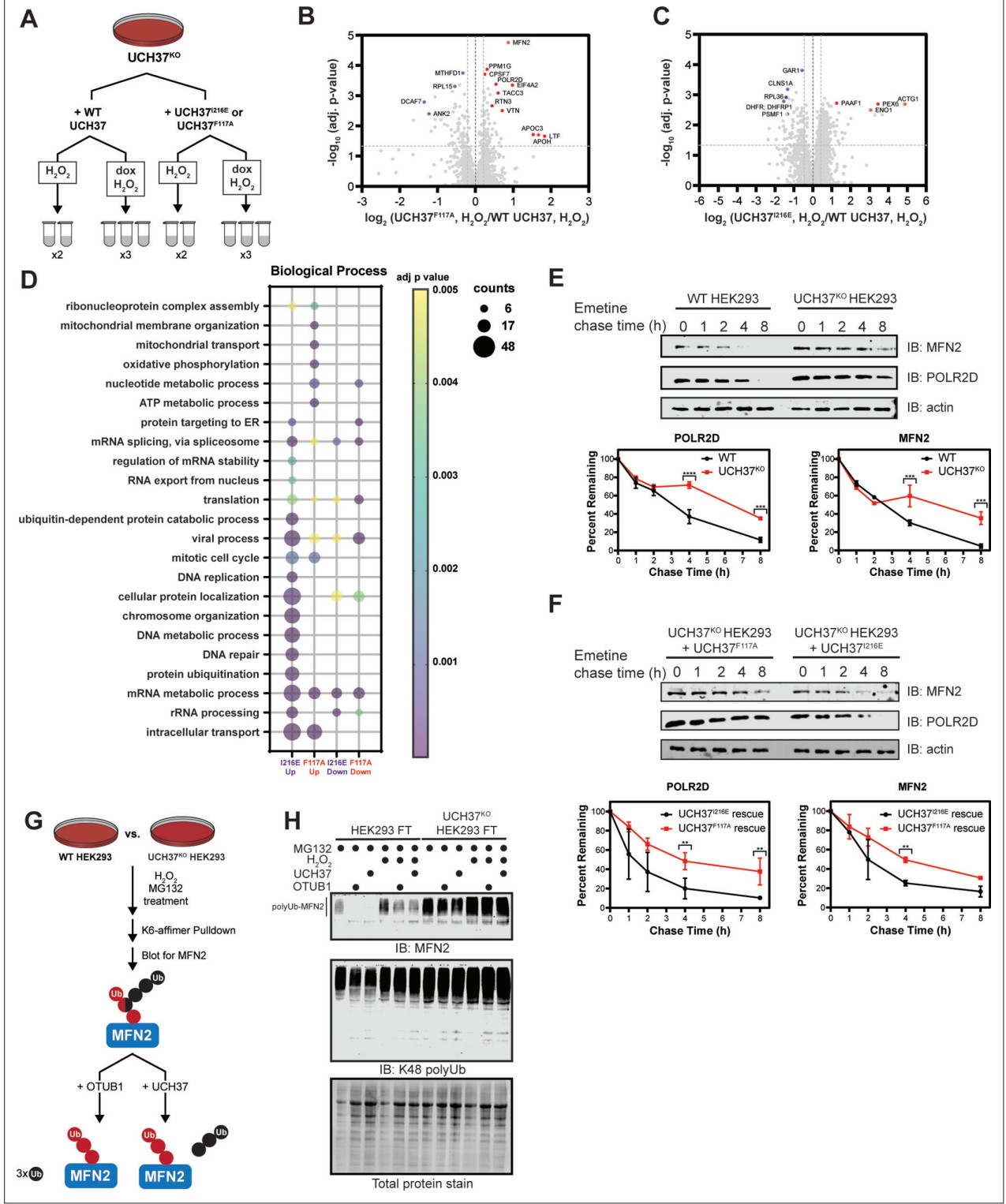

**Figure 7.** Identifying proteins dependent on the K48 chain binding and debranching activity of UCH37 for degradation. (**A**) Scheme for tandem mass tagging (TMT)-based proteomics of UCH37$^{KO}$ cells expressing either wild-type (WT) UCH37, UCH37$^{I216E}$, or UCH37$^{F117A}$. (**B–C**) Volcano plots comparing the proteomes of UCH37$^{F117A}$- and WT UCH37-expressing UCH37$^{KO}$ cells (**B**), and UCH37$^{I216E}$- and WT UCH37-expressing UCH37$^{KO}$ cells. (**C**) In both cases, the cells are treated with H$_2$O$_2$ to induce oxidative stress. (**D**) Biological process analysis of proteins upregulated in UCH37$^{F117A}$- and UCH37$^{I216E}$-expressing UCH37$^{KO}$ cells. (**E**) Western blot analysis of the turnover of POLR2D and mitofusin-2 (MFN2) after shutting off translation with emetine in WT and UCH37$^{KO}$ HEK293 cells. (**F**) Western blot analysis of the turnover of POLR2D and MFN2 after shutting off translation with emetine in UCH37$^{KO}$

*Figure 7 continued on next page*

*Figure 7 continued*

cells reconstituted with either UCH37$^{F117A}$ or UCH37$^{I216E}$. Immunoblotting with α-POLR2D, α-MFN2, and α-actin antibodies. (**G**) Scheme showing the enrichment and analysis K6-linked ubiquitin chains. (**H**) Western blot analysis of K6-affimer-enriched ubiquitinated species from WT and UCH37$^{KO}$ HEK293 cells. The enriched conjugates were treated with OTUB1 (1 μM) or UCH37•RPN13 (2 μM) at rt for 1 hr. Immunoblotting with α-MFN2 and α-K48-linkage specific antibodies.

The online version of this article includes the following source data and figure supplement(s) for figure 7:

**Source data 1.** Tandem mass tagging proteomics analysis of WT, UCH37KO and UCH37 mutants (F117A or I216E) transduced cells.

**Source data 2.** Uncropped gel images of Western blot analysis of emetine chase and K6-affimer enrichment pulldown assay.

**Figure supplement 1.** Identifying proteins dependent on the K48 chain binding and debranching activity of UCH37 for degradation.

reproducibly ~2500 proteins from the 10-plex analysis using an identification cutoff of at least two unique peptides. Protein abundances between replicates show higher correlations (Pearson's correlation coefficient ≥0.91) than those observed between different cell lines (*Figure 7—figure supplement 1C*). Statistical analysis reveals UCH37$^{F117A}$ and UCH37$^{I216E}$ induce significant, but distinct, changes in protein abundance relative to WT (log$_2$ ratio <−0.5 or >0.5, corrected log$_{10}$ p-value >1.3) (*Figure 7B–C*).

Gene ontology (GO) enrichment analysis further illuminates the differences between UCH37$^{F117A}$ and UCH37$^{I216E}$. Proteins that are uniquely upregulated by UCH37$^{F117A}$ participate in mitochondrial membrane organization, mitochondrial transport, oxidative phosphorylation, and nucleotide metabolism (*Figure 7D*). These data suggest that the backside, and thus K48 chain binding and debranching, is important for regulating the levels of proteins related to mitochondrial function and nucleotide metabolism during oxidative stress. By contrast, the frontside appears to regulate the abundance of proteins involved in chromosome organization, DNA metabolism, and DNA repair, as evidenced by an accumulation upon expression of UCH37$^{I216E}$.

To validate these findings, we wanted to examine the effects of UCH37$^{I216E}$ and UCH37$^{F117A}$ on the turnover of specific proteins exhibiting higher abundance upon expression of the latter but not the former. We chose to focus on POLR2D and mitofusin-2 (MFN2). POLR2D (also referred to as Rpb4) regulates various stages of the mRNA life cycle through its interactions with RNA Pol II (*Farago et al., 2003*; *Goler-Baron et al., 2008*; *Harel-Sharvit et al., 2010*). MFN2 is a conserved dynamin-like GTPase localized to the outer membrane of mitochondria, and along with MFN1, regulates mitochondrial dynamics (*Zorzano et al., 2010*). We first compared the levels of POLR2D and MFN2 in WT and UCH37$^{KO}$ cells by immunoblotting and found that both proteins are more abundant in the absence of UCH37 (*Figure 7E*). We then assessed relative turnover rates. In WT cells, POLR2D and MFN2 are rapidly depleted after shutting off translation; however, in UCH37$^{KO}$ cells, both proteins remain relatively stable, indicating that turnover is impaired without UCH37 (*Figure 7E*). Degradation of POLR2D and MFN2 can only be rescued in UCH37$^{KO}$ cells expressing UCH37$^{I216E}$ (*Figure 7F*, *Figure 7—figure supplement 1D*), suggesting the turnover of these proteins is regulated by the K48 chain-specific binding and debranching activity of UCH37.

Lastly, we sought to check whether heterotypic chains bearing K48 linkages accumulate on the putative substrates in the absence of UCH37. MFN2 has previously been shown to be modified with K6-linked chains during mitochondrial stress (*Gersch et al., 2017*). Thus, we used a K6-specific affimer (*Michel et al., 2017*) to isolate polyubiquitinated proteins from WT and UCH37$^{KO}$ cells treated with the proteasome inhibitor MG132 (*Figure 7G*). Western blot analysis of the K6-linked conjugates shows higher levels of polyubiquitinated MFN2 (polyUb-MFN2) and K48 linkages in UCH37$^{KO}$ cells relative to WT (*Figure 7H*). Treatment with H$_2$O$_2$ increases the overall amounts. Addition of the K48 linkage-specific DUB OTUB1 results in the cleavage of Ub conjugates attached to MFN2, indicating the K6 chains also contain K48 linkages, consistent with previous reports (*Michel et al., 2017*). Moreover, the ability of UCH37•RPN13 to hydrolyze polyUb-MFN2 argues that the K6/K48 chains have branchpoints. These data therefore point to specific substrate proteins whose degradation is potentiated by the ability of UCH37 to debranch K48 Ub chains.

## Discussion

Here, we describe how UCH37—a DUB with fundamental roles in chromatin biology and protein degradation—achieves specificity for K48 Ub chains. The surprising finding is that the cS1 site, which is thought to be the principal Ub-binding site, is dispensable for K48-linkage-specific binding. Specificity is instead accomplished by avid binding to the opposite face of the enzyme relative to the S1 site. Mutational analysis shows the backside is not only required for K48 chain binding but also chain debranching and the ability of UCH37 to promote proteasomal degradation. Defects in the cS1 site and backside have distinct effects on the turnover of a subset of proteins, enabling the identification of branched chain-modified substrates whose degradation depends on UCH37. These results raise an intriguing question of whether some DUBs use more than one S1 site to perform different biochemical activities.

### A model for chain debranching

Considering the mechanisms by which some USP DUBs target both Ub chains and ubiquitinated substrates, the most logical model for the different activities of UCH37 would invoke a single S1 site (*Figure 1A*). USP30, for example, is thought to use the same S1 site to cleave K6-linked Ub chains and remove Ub from numerous mitochondrial outer membrane proteins (*Gersch et al., 2017*). Chain type specificity and substrate promiscuity are both possible due to an accessible active site geometry and the presence of a second Ub-binding site, termed the S1′ site. A similar scenario can be envisioned for UCH37. When the substrate is a C-terminal adduct or an N-terminally ubiquitinated protein, the Ub moiety interacts with the cS1 site and the scissile peptide bond enters the active site because the flexible CL moves out of the way. By contrast, when the substrate is a branched Ub chain, the distal moiety binds the S1 site and the proximal Ub interacts with a portion of the CL only if a K48 linkage is present between the two subunits (*Figure 1B*).

Our data, however, suggest that in addition to the cS1 site there is another S1 site located on the backside (termed noncanonical S1 or ncS1 site) dedicated to binding and catalyzing K48 chain debranching (*Figure 8A*). Whether the α5–6 motif or the region containing the CL and L181 serves as the ncS1 site depends on the orientation of the K48 chain most conducive to catalysis (*Figure 8B*). Unbranched K48 chains, which are generally poor substrates, appear to preferentially bind the proximal Ub at the α5–6 motif. In this orientation, molecular docking analysis suggests the isopeptide bond would be sterically shielded from the catalytic Cys (*Figure 4H*; pose 1). An arrangement that might be more catalytically productive is one in which the two subunits swap positions. With the proximal Ub near the CL and L181, the K48 distal Ub can still interact with the α5–6 motif (*Figure 8B*), but the scissile bond could be in a better position for cleavage as the docking model shows that the only residue blocking access to C88 is Q82 (*Figure 4H*; pose 2).

How the third Ub that branches off from the proximal subunit facilitates isopeptidase activity remains an open question. One possibility is that it provides enough steric bulk to shift the equilibrium in favor of the proximal subunit binding near the CL and L181, which would allow the isopeptide bond to move closer to the catalytic Cys. However, this model alone does not explain why Ub must be at all three positions of a branchpoint, as we have previously shown (*Deol et al., 2020*), nor does it explain the enhanced selectivity and debranching activity conferred by RPN13. Another scenario, more consistent with the data, is one in which the non-K48 distal Ub forms weak interactions with the frontside of the enzyme, for example, the 211–217 loop. The precise orientation is unknown but considering the I216E substitution does not affect the rate of debranching, the nature of the interaction between the non-K48 distal Ub and the frontside is unlikely to resemble mono-Ub binding the cS1 site (*Figure 8B*). With the non-K48 distal Ub on the frontside, the proximal Ub would be forced to reside near the CL and L181 and the K48 distal Ub would be bound to the α5–6 motif. Cleavage of the branched chain would mean the α5–6 motif acts as the ncS1 site. Thus, UCH37 could potentially use distinct S1 sites for different biochemical functions. Future studies will focus on elucidating the precise mechanism by which UCH37 together with RPN13 discerns between unbranched and branched chains.

### Substrates requiring the debranching activity of UCH37 for degradation

The different biochemical activities of UCH37 present challenges in pinpointing which proteins depend on the K48 chain debranching activity for degradation. This issue can be addressed with mutants that

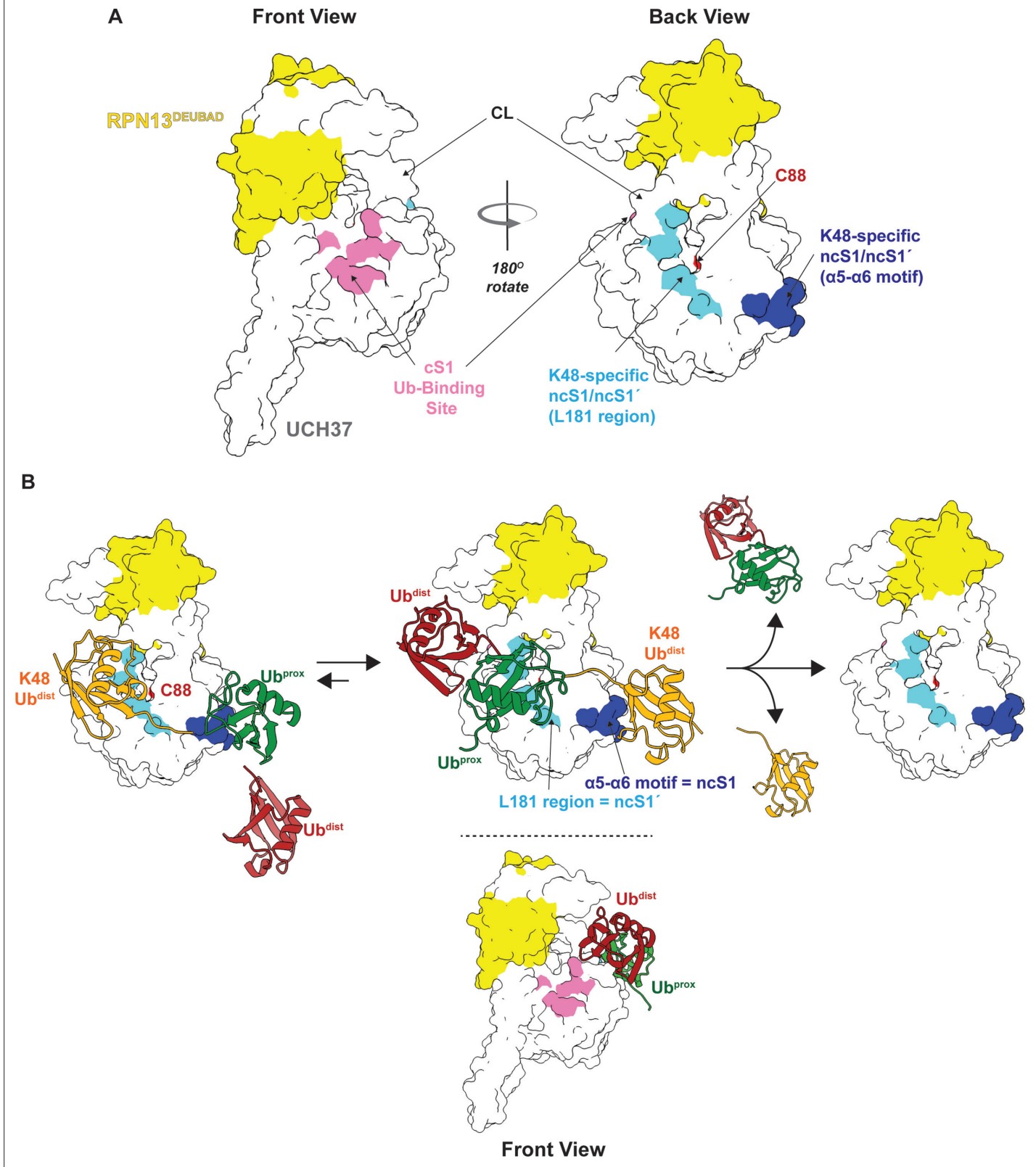

**Figure 8.** Proposed debranching model. (**A**) Surface depiction of UCH37 showing the canonical S1 (cS1) ubiquitin (Ub)-binding site and the new K48-specific binding sites identified in this study. (**B**) Proposed mechanism for chain debranching using the K48-specific binding sites. The K48-linked portion of a branched chain engages the K48-specific Ub-binding sites in two different orientations: one with the proximal Ub (Ub^prox) docked at the α5–6 motif and the other with the distal Ub (Ub^dist) bound to that site. As the docking models show (*Figure 4H*), the K48 isopeptide bond is less obstructed and

*Figure 8 continued on next page*

*Figure 8 continued*

closer to the catalytic C88 residue when the K48 Ub$^{dist}$ moiety is bound to α5–6 and Ub$^{prox}$ is bound near the L181 region. In this orientation, the other Ub$^{dist}$ at the branchpoint is positioned near the frontside of the enzyme. With K48 isopeptide bond cleavage occurring on the backside, the α5–6 motif would serve as the noncanonical S1 (ncS1) site and the L181 region would be the ncS1´ site.

specifically disrupt K48 chain binding and debranching without altering hydrolytic activity, that is, F117A and F121A. By comparing the effects of UCH37$^{F117A}$, UCH37$^{F121A}$, and UCH37$^{I216E}$, on total protein abundance we found that the mitochondrial fusion protein MFN2 and the RNA polymerase II subunit POLR2D depend on the K48 chain binding and debranching activity of UCH37 for degradation. Previous studies have shown that MFN2 is polyubiquitinated and degraded by the proteasome in response to cellular stress. The E3 ligase HUWE1 modifies MFN2 with Ub chains bearing a mixture of K6, K11, and K48 linkages (*Leboucher et al., 2012*; *Michel et al., 2017*; *Senyilmaz et al., 2015*). Our results are not only consistent with these findings, but also provide direct evidence that the chains attached to MFN2 contain branchpoints that are removed by UCH37 to facilitate degradation.

Why chain debranching is required for the efficient removal of MFN2, or any other protein, by the proteasome is less clear. There could be several reasons. For example, the removal of branchpoints could assist in the transfer of a polyubiquitinated protein from an effector protein to the proteasome. The proteasomal shuttle protein HHR23A contains two Ub-binding UBA domains, one with a preference for K48 linkages and the other displaying no linkage selectivity (*Sims et al., 2009*; *Varadan et al., 2005*). Coordination between the two UBA domains could enable high affinity binding to a K48 branched chain (*Haakonsen and Rape, 2019*). This would prioritize branched chain-modified substrates for degradation, but debranching would have to occur to allow the proteasome to takeover. A similar scenario can be envisioned for the transfer of substrates between the AAA + ATPase p97/VCP and the proteasome, as p97 is required for the degradation of MFN2 (*Tanaka et al., 2010*) and been shown to act as an effector of branched chains (*Yau et al., 2017*). In our in vitro degradation assays, however, these shuttling factors are largely absent.

The model we favor is one in which debranching facilitates clearance of chains removed by the intrinsic proteasomal DUB RPN11 after each round of degradation. Unfolding and translocating Ub chains through the pore of the proteasome present a major roadblock for the degradation process (*Worden et al., 2017*). Cotranslocational removal of Ub modifications by the Zn$^{2+}$-dependent JAMM/MPN DUB RPN11 (*Verma et al., 2002*; *Yao and Cohen, 2002*), which resides directly above the entrance to the pore of the 19 S AAA + ATPases (*Dambacher et al., 2016*), alleviates this issue. However, the Ub chains that are cleaved en bloc by RPN11 can remain bound to the proteasome and inhibit additional rounds of degradation. This scenario could be more acute with branched chains compared to their unbranched counterparts considering the 19 S subunit RPN1 strongly prefers binding K11/K48 branched chains (*Boughton et al., 2020*). Debranching by UCH37 could then attenuate Ub chain binding, freeing the Ub receptors of the proteasome for another round of degradation.

## Implications for UCH37-catalyzed debranching in the INO80 complex

Structural studies have elegantly shown that the DEUBAD domain of the INO80 subunit NFRKB/INO80G (INO80G$^{DEUBAD}$) inhibits binding of Ub to UCH37 by obscuring elements comprising the cS1 Ub-binding site (*Sahtoe et al., 2015*; *Vander Linden et al., 2015*). The principal driver of the blockade is the interaction between the L38 pocket of the S1 site and the FRF loop of INO80G$^{DEUBAD}$. What is also striking about the structure of UCH37•INO80G$^{DEUBAD}$, is the extended helix α6 of INO80G$^{DEUBAD}$ that drapes over the backside of the enzyme. Y142 in α6 forms direct contacts with L181 on the backside of UCH37. According to our data, L181 comprises a backside Ub-binding site. Thus, in addition to the cS1 site, key residues of the K48 chain-specific binding site could be blocked by INO80G$^{DEUBAD}$. This suggests the INO80 complex could potentially inactivate all activities of UCH37. That said, the INO80 complex is instrumental in many of the same biological processes upregulated in knockout cells expressing the UCH37$^{I216E}$ mutant (*Poli et al., 2017*). Thus, it is plausible that UCH37 functions primarily through the cS1 site in the context of entire INO80 complex.

## Conclusion

In summary, our data uncover an entirely unique region of UCH37 required for K48 chain specificity, chain debranching, and promoting the proteasomal degradation of branched chain-modified proteins.

What is particularly exciting is the prospect that a DUB can use different S1 sites to target Ub chains and ubiquitinated substrates, suggesting inhibitors could be designed to block one activity but not the other. For instance, a small molecule that specifically blocks Ub chain binding and debranching could serve as a powerful tool to probe the role of UCH37-catalyzed debranching in many biological paradigms.

Our findings could also be relevant to other UCH family members. The tumor suppressor BAP1 (*Carbone et al., 2013*), which has been implicated in various cellular processes, shares 42% sequence identity with the CD of UCH37. The aromatic residues forming the nexus of UCH37's K48 chain-specific binding and debranching activities are also conserved in BAP1 (F118 and F122). Due to the ability to deubiquitinate histone H2AK119ub (*Sahtoe et al., 2016*; *Scheuermann et al., 2010*), most studies have focused on the nuclear activities of BAP1. Much less is known about BAP1's cytoplasmic activities (*Szczepanski and Wang, 2021*). Using the backside aromatic-rich motif, it is possible that cytoplasmic BAP1 acts in a K48 chain editing capacity, with target specificity dictated by many of its interacting partners.

## Materials and methods

*All materials written in the following format: REAGENT/ RESOURCE (Source, Identifier).

### Antibodies

Anti UCH37 (Abcam, Cat. # ab124931)
Anti RPN11 (Abcam, Cat. # ab109130)
Anti RPN13 (Cell Signaling Technology, Cat. # D9Z1U)
Anti RPT2 (Abcam, Cat. # ab3317)
Anti PSMB7 (R&D Systems, Cat. #MAB7590)
Anti USP14 (Abcam, Cat. # ab56210)
Anti Ub (P4D1, Enzo Lifesciences, Cat. # BML-PW0930)
Anti Mitousin2 (Cell Signaling Technology, Cat. # 9,482 S)
Anti POLR2D (Proteintech, Cat. # 16093–1-AP)
Anti RPB (CTD) (Cell Signaling Technology, Cat. #2629)
Anti β-actin (Cell Signaling Technology, Cat. # 3700T)
Anti K48-linkage Specific (Cell Signaling Technology, Cat. # D9D5)
Goat Anti Mouse IR Dye 800CW (LI-COR Biosciences, Cat. # 926–32210)
Goat Anti Rabbit IR Dye 680RD (LI-COR Biosciences, Cat. # 926–68071)
Goat Anti Rabbit IR Dye 800CW (LI-COR Biosciences, Cat. # 926–32211)

### Bacterial and viral strains

Rosetta 2(DE3)pLysS (EMD Millipore Novagen, Cat. # 71403–3)
BL21(DE3)pLysS (Promega, Cat. # L1191)
One Shot Stbl3 (Fisher Scientific, Cat. # C737303)

### Chemicals, peptides, and recombinant proteins

ProBlock Gold Mammalian Protease Inhibitor Cocktail (Gold Biotechnology, Cat. # GB-331–5)
Ammonium Chloride ($^{15}$N, 99%) (Cambridge Isotope Laboratories, Cat. # NLM-467)
Sypro Ruby Stain (Fisher Scientific, Cat. # S12000)
Cy5 maleimide (Lumiprobe, Cat. #23080)
DBCO-Cy5 (Sigma, Cat. # 777374)
Click-iT AHA (L-Azidohomoalanine) (Fisher Scientific, Cat. #C10102)
Trypsin (Promega, Cat. # V5113)
Formic Acid (Sigma-Aldrich, Cat. # 399388)
Acetic Acid (Fisher Scientific, Cat. # 351269–4)
Creatine phosphate disodium salt (Abcam, Cat. # ab146255)
Creatine Kinase (Sigma-Aldrich, Cat. # 10127566001)
Adenosine-5'-triphosphate (Goldbio, Cat. # A-081–5)
Ub-AMC (Boston Biochem, Cat. # U-550)

Suc-LLVY-AMC (Boston Biochem, Cat. # S-280)
MG132 (Fisher Scientific, Cat. # 508339)
Bortezimib (Selleck Chemicals, Cat. # S1013)
Polybrene (Sigma, Cat. # TR-1003-G)
Lipofectamine 3000 (Fisher Scientific, Cat. # L3000008)
Doxycycline hyclate (Sigma, Cat. # D9891)
L-Methionine (Sigma, Cat. #64319–25 G-F)
Carfilzomib (PR-171) (Selleck Chemicals, Cat. # S2853)
MTS-Alkynyldiazirine (Redbrick Molecular, Cat. # RBM-0000766)
Dithiothreitol (Gold Biotechnology, Cat. # DTT10)
Dimethyl sulfoxide (Alfa Aesar, Cat. # 42780)
Iodoacetamide (Sigma-Aldrich, Cat. # I1149)
Acetonitrile (Fisher Scientific, Cat. # A955-4)
D-Glucose (Cambridge Isotope Laboratories, Cat. # DLM-2062–10)
EZ-Link Iodoacetyl-PEG2-Biotin (Fisher Scientific, Cat. # PI21334)

## Critical commercial assays

pENTR/SD/D-TOPO Cloning Kit (Fisher Scientific, Cat. # K242020)
Gateway LR Clonase II Enzyme Mix (Fisher Scientific, Cat. # 11-791-020)

## Experimental models: cell lines

HEK293 Expressing Rpn11-HTBH (Applied Biological Materials, T6007)
HEK293 Expressing Rpn11-HTBH RPN13 KO (*Deol et al., 2020*)
HEK293 FT (ATCC, CRL-3216)

HEK293 FT UCH37[KO] (*Deol et al., 2020*)
HEK293 GFP[u] (ATCC, CRL-2794)
HEK293 GFP[u] UCH37[KO] (*Deol et al., 2020*)

## Recombinant DNA (Source, Identifier)

pMCSG20: NleL (aa 170–782) (*Valkevich et al., 2014*, N/A)
pVP16: UCH37 (DNASU, HsCD00084019)
pET19: RPN13 or Adrm1 (Addgene, Plasmid #19423)
pET28b: E1 (*Trang et al., 2012*; N/A)
pGEX-4T2: UBE2D3 (*Valkevich et al., 2014*, N/A)
pGEX-6P1: UBE2S-UBD (Addgene, Plasmid #66713)
pGEX-6P1: AMSH (aa 219–424) (*Trang et al., 2012*, N/A)
pOPINK: OTUD1 (Addgene, Plasmid #61405)
pVP16: OTUB1 (*Pham et al., 2016*, N/A)
pOPINS: UBE3C (Addgene, Plasmid #66711)
pDEST17: UBE2R1 (*Pham et al., 2016*, N/A)
pST39: UBE2N/UBE2V2 (*Pham et al., 2016*, N/A)
pET28b: UBC1 (DNASU, ScCD00009212)
pOPINK: RSP5 (DNASU, ScCD00008707)
pOPINS: Titin I27[V15P] 23 K-35 (*Bard et al., 2019*, N/A)
pET22b: Ub and Ub variants (*Trang et al., 2012*, N/A) pET22b: GFP (Addgene, Plasmid #11938)
pMD2.G (Addgene, Plasmid #12259)
psPAX2 (Addgene, Plasmid #12260)
pINDUCER21 (Addgene, Plasmid #46948)
pCW57-MCS1-P2A-MCS2 (RFP) (Addgene, Plasmid #80923)

## Software and algorithms

Typhoon FLA 9500 (GE Healthcare)
Odyssey CLx Imager (LICOR)
Image Studio software (LICOR Biosciences)

Prism 9 (Graphpad Software)
Pinpoint 1.4 (Thermo Fisher Scientific)
Proteome Discoverer 2.3 (Thermo Fisher Scientific)
FlowJo 10.4 (FlowJo, LLC)
ImageJ/FIJI (*Schindelin et al., 2012*)
ATSAS Software Suite
Primus (*Konarev et al., 2003*)
ASTRA Software 6.0 (Wyatt Technology)
Xcalibur Software (Thermo Fisher Scientific)
UCSF Chimera (University of California RBVI, San Francisco)
GROMACS 2018 package (KTH Royal Institute of Technology, Uppsala University, University of Groningen)
HADDOCK 2.2 (Webserver Utrecht University, Bonvin Lab)
FoXS webserver (University of California, San Francisco, Sali Lab)
SWISS-MODEL (Swiss Institute of Bioinformatics)

## Other

His60 Ni Superflow resin (Clontech, Cat. # 635660)
Glutathione resin (GenScript, Cat. # L00206)
Amylose resin (NEB, Cat. # E8021S)
Streptavidin resin (GenScript, Cat. # L00353)
Slide-A-lyzer MINI dialysis units (3.5 kDa MWCO) (Thermo Scientific, Cat. # PI69552)
100 mg SEP-PAK $C_{18}$ column (Waters, Cat. # wat043395)
C18 StageTips (Thermo Scientific, Cat. # SP301)
Zeba Spin Desalting Column (Thermo Scientific, Cat. # 89889)
NuPAGE Novex 12% Bis-Tris Protein Gels (Fisher Scientific, Cat. # NPO343BOX)
4–20% Mini-PROTEAN Gels (Bio-Rad, Cat. # 4561096)
Syringe Filters, PES, 0.45 µm (Genesee Scientific, Cat. # 25–246)
PD-10 Desalting Columns (GE Healthcare)
OmniCure S1500 Spot UV Curing System (Excelitas Technologies)
NuPAGE 12% Bis-Tris Mini Protein Gel (Thermo Scientific)
Sep-Pak C18 Cartridge (Waters)

## Human cells

HEK293 cells stably expressing His-biotin affinity tagged human RPN11 (RPN11-HTBH) (*Wang et al., 2007*), HEK293FT, and HEK293 GFP[u] cells were cultured at 37°C under 5% $CO_2$ using high glucose Dulbecco's Modified Eagle Medium (DMEM) supplemented with 10% Fetal Bovine Serum (FBS), 1x Glutamax (Gibco), and 1xPen/Strep.

## Generation of UCH37 variants

Human UCH37 (isoform3) in pVP16 plasmid was purchased from DNASU. Mutations in the sequence encoding UCH37 were generated using site-directed mutagenesis following the QuikChange protocols using Phusion polymerases (New England Biolab). The variant of UCH37 with the ULD removed (UCH37[CD]) was generated using restriction-enzyme free cloning protocols from integrated DNA technology using Phusion polymerases (New England Biolab).

## Expression and purification of Ub conjugation machinery

E1, UBE2D3, UBE2R1, UBE2N/UBE2V2, UBC1, and UBE3C were purified as previously described with slight modification (*Bashore et al., 2015*; *Deol et al., 2020*; *Michel et al., 2015*; *Pham et al., 2016*; *Trang et al., 2012*). Briefly, E1, UBE2D3, UBE2R1, and UBE2N/UBE2V2 constructs were expressed in Rosetta 2(DE3)pLysS *Escherichia coli* cells grown at 37°C in LB media supplemented with 40 µg/mL ampicillin. Once cells reached an $OD_{600}$ of 0.6–0.8, 400 µM Isopropyl ß-D-1-thiogalactopyranoside (IPTG) was added, and the temperature was reduced to 20°C to induce protein expression. Cells were then harvested after 16 hr and resuspended in Ni-NTA lysis buffer (50 mM Tris pH 7.5, 300 mM NaCl, 1 mM EDTA and 10 mM imidazole), lysed by sonication, and clarified by centrifugation. Clarified lysate

was incubated with Ni-NTA resin for 2 hr at 4°C, washed with Ni-NTA lysis buffer, and eluted into Ni-NTA elution buffer (Ni-NTA lysis buffer plus 300 mM imidazole).

NleL (aa 170–782) was purified as previously described (*Deol et al., 2020*; *Valkevich et al., 2014*), but with a slight modification. Briefly, NleL was expressed in BL21(DE3)pLysS *E. coli* cells grown at 37°C in LB media supplemented with 40 µg/mL ampicillin. Once cells reached an $OD_{600}$ of 0.6, IPTG (200 µM) was added, and the temperature was reduced to 20°C to induce protein expression. Cells were then harvested after 16 hr and resuspended in GST lysis buffer (50 mM Tris pH 8.0, 200 mM NaCl, 1 mM EDTA and 1 mM DTT), lysed by sonication, and clarified by centrifugation. Clarified lysate was then incubated with GST resin for 2 hr at 4°C, washed with GST lysis buffer, and eluted into GST elution buffer (GST lysis buffer plus 10 mM reduced glutathione). Eluate was concentrated in TEV protease buffer (50 mM Tris pH 8.0, and 0.5 mM TCEP), cleaved overnight with TEV protease, and further purified using anion exchange chromatography.

## Synthesis of defined Ub chains

Native homotypic chains were synthesized as previously described (*Deol et al., 2020*). Briefly, 2 mM Ub, 300 nM E1, 3 µM UBE2R1 (K48), and 3 µM UBE2N/UBE2V2 (K63) were mixed in Ub chain synthesis reaction buffer (20 mM ATP, 10 mM $MgCl_2$, 40 mM Tris-HCl pH 7.5, 50 mM NaCl, and 6 mM DTT) overnight at 37°C. Native K6/48 branched trimer was synthesized by mixing 2 mM K6/48 R Ub, 1 mM UbD77, 300 nM E1, 10 µM UBE2D3, and 1 µM NleL in ubiquitin chain synthesis reaction buffer overnight at 37°C.

All reactions for native chains were quenched by lowering the pH with the addition of 5 M ammonium acetate pH 4.4. Enzymes were then precipitated through multiple freeze-thaw cycles and further purified using cation exchange chromatography.

## Generation of high molecular weight Ub chains

Native K6/K48 and K48/K63 HMW Ub chains were synthesized as previously described (*Deol et al., 2020*). Briefly, *K6/K48 Ub chains* were synthesized by mixing 1 mM Ub, 150 nM E1, 5 µM UBE2D3, and 3 µM NleL in Ub chain synthesis reaction buffer and *K48/K63 Ub chains* were synthesized using 1 mM Ub, 150 nM E1, 5 µM UBE2R1, and 5 µM UBE2N/UBE2V2 in Ub chain synthesis reaction buffer. Reaction mixtures were all incubated at 37°C overnight with shaking. To synthesize biotinylated K6/48 HMW Ub chains, 100 µM Ub K63C was mixed with 1 mM EZ-link iodoacetyl-PEG2-biotin (ThermoFisher Scientific) in 10% Dimethyl sulfoxide (DMSO) at in biotinylation reaction buffer (50 mM HEPES pH 7.5, 5 mM $MgCl_2$, 25 mM KCl, and 1 mM TCEP) and incubated at 37°C overnight. All Ub chains were purified using size exclusion chromatography (Superdex 75) to isolate HMW chains >35 kDa.

## Protein expression and purification

Complexes containing UCH37 and RPN13 or RPN13[DEUBAD] were expressed and purified as previously described (*Deol et al., 2020*), but with a slight modification. Briefly, UCH37 and RPN13/RPN13[DEUBAD] constructs were expressed in BL21(DE3)pLysS *E. coli* cells grown at 37°C in LB media supplemented with 40 µg/mL ampicillin. Once cells reached an $OD_{600}$ of 0.6, IPTG (400 µM) was added, and the temperature was reduced to 20°C. Cultures were harvested after 16 hr and frozen at −80°C with UCH37•RPN13 or UCH37•RPN13[DEUBAD] complexes mixed 1:1 (volume) prior to lysis. Cell pellets were resuspended in amylose lysis buffer (50 mM HEPES pH 7.4, 150 mM NaCl, 1 mM EDTA, and 1 mM TCEP), lysed by sonication, and clarified by centrifugation. UCH37•RPN13 and UCH37•RPN13[DEUBAD] were purified following different chromatographic steps. For UCH37•RPN13, clarified lysate was incubated with amylose resin for 2 hr at 4 °C, washed with amylose lysis buffer, and eluted into amylose elution buffer (lysis buffer plus 10 mM maltose), and incubated overnight with TEV protease at 4°C. Then TEV protease cleaved product was incubated with Ni-NTA resin for 1 hr, washed with Ni-NTA lysis buffer, and eluted with Ni-NTA elution buffer (Ni-NTA lysis buffer plus 300 mM imidazole). For UCH37•RPN13[DEUBAD], clarified lysate was incubated with Ni-NTA resin for 2 hr at 4°C, washed with Ni-NTA lysis buffer, and eluted into Ni-NTA elution buffer, and then incubated overnight with TEV protease. The TEV protease cleaved product was then incubated with GST resin for 1 hr, washed with lysis buffer, and eluted with GST elution buffer (GST lysis buffer plus 10 mM glutathione). Eluate were incubated overnight with 3 C protease overnight at 4°C. For both UCH37•RPN13 and UCH37•RPN13[DEUBAD], eluate was concentrated and loaded onto a Superdex 200 (GE) gel filtration column in

gel filtration buffer (50 mM HEPES pH 7.5, 50 mM NaCl, 1 mM EDTA, and 1 mM TCEP). For UCH37 and UCH37deltaULD, clarified lysate was subjected to Ni-NTA chromatography followed by further purification using anion exchange chromatography and Superdex 75 (GE) gel filtration column in gel filtration buffer.

## Proteasome (Ptsm) purification

Proteasomes (Ptsms) were purified as previously described (*Deol et al., 2020*). WT and RPN13[KO] cells stably expressing RPN11-HTBH were grown, harvested, and lysed in Ptsm lysis buffer (40 mM HEPES pH 7.4, 40 mM NaCl, 10 mM MgCl$_2$, 2 mM ATP, 1 mM DTT, and 10% glycerol). The lysates were clarified at 20,000× g for 20 min, and the supernatant was incubated with streptavidin resin overnight with rocking at 4°C. The resin was washed and incubated for 10 min intervals with high salt wash buffer (Ptsm buffer containing 200 mM NaCl) on ice with rocking. The resin was then resuspended in low salt wash buffer (Ptsm lysis buffer without DTT) and incubated with TEV protease for 1.5 hr at room temperature. Minor modifications were made for the purification of Ptsm complexes containing variants of UCH37•RPN13, that is, I216E, F117A, and F121A. Clarified lysates derived from RPN13[KO] cells were incubated with streptavidin resin overnight with rocking. The resin was pelleted, resuspended in Ptsm buffer containing 10 µM WT or mutant recombinant UCH37•RPN13 complexes, and further incubated for 4 hr with rocking prior to high salt washes. The resulting Ptsm complexes were characterized by western blot analysis, Ub-AMC, and suc-LLVY-AMC activity.

## Purification, ubiquitination and fluorescent labeling of titin I27[V15P]23-K-35 and UBE2S-UBD

Titin I27[V15P]23-K-35 was purified as previously described (*Deol et al., 2020*). Briefly, titin I27[V15P] 23 K-35 was expressed in BL21(DE3)pLysS *E. coli* cells grown at 30°C in 2× YT media containing 1% glycerol and supplemented with 40 µg/mL ampicillin. When the OD$_{600}$ reached 1.2–1.5, cells were induced with IPTG (400 µM), and grown for an additional 5 hr at 30°C. Cultures were harvested, flash frozen, resuspended in titin lysis buffer (60 mM HEPES pH 7.5, 100 mM NaCl, 100 mM KCl, 10 mM MgCl$_2$, 0.5 mM EDTA, 1 mg/mL lysozyme, 2 mM PMSF, 20 mM imidazole, and 10% glycerol), lysed by sonication, and clarified by centrifugation. Clarified lysate was then incubated with Ni-NTA resin for 2 hr at 4°C, washed with Ni-NTA binding buffer (50 mM HEPES pH 7.5, 150 mM NaCl, and 20 mM imidazole), and eluted with elution buffer (binding buffer plus 300 mM imidazole). Eluate was buffer exchanged into Ulp1 protease buffer (50 mM HEPES pH 7.5 and 150 mM NaCl), cleaved overnight with Ulp1, and further purified using a Superdex 200 (GE) gel filtration column in 50 mM HEPES pH 7.5 and 5% glycerol. 100 µM purified titin was ubiquitinated by mixing with 5 µM E1, 5 µM Ubc1, 20 µM Rsp5, and 2 mM Ub in labeling buffer (60 mM HEPES pH 7.5, 20 mM NaCl, 20 mM KCl, 10 mM MgCl$_2$, and 2.5% glycerol) containing 1× ATP regeneration mix for 3 hr at room temperature followed by the addition of 5 µM UBE2R1 and incubation overnight at 4°C.

UBE2S-UBD was purified as previously described. Briefly, UBE2S-UBD constructs were expressed in Rosetta 2(DE3)pLysS *E. coli* cells grown at 37°C in LB media supplemented with 40 µg/mL ampicillin. Once the OD$_{600}$ reached 0.6–0.8, IPTG (400 µM) was added, and the temperature was reduced to 16°C. After 16 hr of expression, cultures were harvested, resuspended in UBE2S lysis buffer (270 mM sucrose, 50 mM Tris pH 8.0, 50 mM NaF, and 1 mM DTT), lysed by sonication, and clarified by centrifugation. Clarified lysate was then incubated with GST resin for 2 hr at 4°C, washed with washing buffer (25 mM Tris pH 8.0, 150 mM NaCl, and 5 mM DTT), and eluted in elution buffer (washing buffer plus 10 mM glutathione). Eluate was buffer exchanged into 3 C protease buffer (50 mM Tris pH 8.0 and 150 mM NaCl), and cleavage was performed with HRV 3 C protease overnight, followed by further purification by size exclusion chromatography. UBE2S-UBD was ubiquitinated by mixing 0.6 µM Ub, 150 nM E1, and 5 µM UBE2S-UBD in reaction buffer (10 mM ATP, 10 mM MgCl$_2$, 40 mM Tris pH 8.5, 100 mM NaCl, 0.6 mM DTT, and 10% (v/v) glycerol) for 3 hr at 37°C, followed by addition of 3 µM AMSH and 0.5 µM OTUD1. The resulting mixture was incubated overnight at 37°C. An additional amount of AMSH and OTUD1 was added, and the mixture was incubated for 3 hr at 37°C prior to the purification by size exclusion chromatography to isolate products with a molecular weight >35 kDa. Purified HMW K11-linked chains were then mixed with 0.6 µM Ub, 150 nM E1, and 3 µM UBE2R1 in reaction buffer (10 mM ATP, 10 mM MgCl$_2$, 40 mM Tris pH 8.5, 100 mM NaCl, 0.6 mM DTT, and 10%

(v/v) glycerol) at 37°C overnight, followed by purification using size exclusion chromatography to collect >35 kDa products.

2 mg/mL ubiquitinated titin or UBE2S-UBD was fluorescently labeled using Cy5-maleimide at pH 7.2 for 2 hr at room temperature and quenched with excess DTT. Free dye was separated from the substrate using a Zeba spin desalting column and buffer exchanged into labeling buffer (60 mM HEPES pH 7.5, 20 mM NaCl, 20 mM KCl, 10 mM MgCl$_2$, and 2.5% glycerol).

## Ub chain debranching assay and Ptsm-mediated degradation

The debranching and degradation assays were performed as previously described (*Deol et al., 2020*), but with slight modifications. For the debranching assays, a stock solution of recombinant UCH37 or replenished Ptsms was warmed to 37°C in DUB buffer or Ptsm buffer along with either unlabeled HMW Ub chains (0.2 mg/mL, for UCH37) or biotinylated HMW Ub chains (0.25 mg/mL, for Ptsm). Reactions were initiated by the addition of UCH37 or Ptsm and quenched with 6× Laemmli loading buffer after 1 or 2 hr. Samples were then separated on a 15% SDS-PAGE gel. Gels were stained with Sypro Ruby to image unlabeled HMW Ub chains. Biotinylated HMW Ub chains were imaged by western blot analysis using streptavidin Alexa-Fluorophore 647.

For the degradation assays, ubiquitinated substrates (250 ng/µL) and Ptsms (200 ng//µL) were warmed to 37°C in Ptsm degradation reaction buffer (40 mM HEPES pH 7.4, 150 mM NaCl, MgCl$_2$) supplemented with 1× ATP-regeneration mix. Reactions were incubated for 1 hr at 37°C and quenched with 6× Laemmli loading buffer. Samples were loaded onto 10–16% tricine gels, separated using SDS-PAGE, and imaged using Cy5 fluorescence on a Typhoon FLA 9500 (GE) set at a pixel density of 50 µm/pixel.

## Fluorescence polarization assay

Catalytically inactive UCH37[C88A]•RPN13 was serially diluted in fluorescence polarization (FP) buffer (50 mM Tris-HCl pH 7.5, 150 mM NaCl, 1 mM DTT, 0.01% Brij-35) and loaded into a 384-well, black nonbinding polystyrene plate (Corning). A fixed concentration of Ub-fluorescein (50 nM) was then added to each well. After incubating for 1 hr at 25°C, FP was measured using a plate reader (Biotek Snergy 2) with polarized filters and optical modules for fluorescein ($\lambda_{ex}$ = 480 nm, $\lambda_{em}$ = 535 nm). FP values (mP) were calculated from raw parallel and perpendicular fluorescence intensities. Graphpad Prism 8.0 was used to fit the data, and the equilibrium dissociation constant $K_d$ was calculated as previously described.

## Steady-state kinetic analysis with defined Ub chains

Stock solutions of enzymes and Ub chains were prepared in DUB buffer (50 mM HEPES pH 7.5, 50 mM NaCl, and 2 mM DTT). All reactions were performed at 37°C. Each sample along with a Ub and di-Ub standard was then separated on a 15% SDS-PAGE gel and stained with Sypro Ruby. Gels were visualized on a Typhoon FLA 9500 (GE), and densitometry was performed using Image Studio. Initial velocities of Ub and di-Ub formation were converted to concentration per minute (µM/min). These values were then fit to the Michealis-Menten equation using nonlinear regression in Graphpad Prism 8.0. Error bars represent the standard deviation of three biological replicates for each reaction performed using UCH37•RPN13 complexes.

## Size exclusion chromatography coupled with multiangle light scattering (SEC-MALS)

The molecular masses of UCH37 complexes with Ub variants were determined using online SEC-MALS. Samples (1 mg/mL) were injected in a total volume of 100 µl onto a 7.8 mm I.D. × 30 cm, 5 µm TSKgel column (TOSOH) equilibrated in 20 mM HEPES pH 7.4% and 10% isopropanol, at a flow rate of 0.5 ml/min. The weight-averaged molecular mass of material contained in chromatographic peaks was calculated using the ASTRA software version 6.0 (Wyatt Technology Corp). An overall average molecular mass and polydispersity term for each species were calculated by combining and averaging the results from the individual measurements at each 0.5 s time interval of the selected peak.

## Isothermal titration calorimetry analysis

ITC measurements were performed on a MicroCal Auto-ITC200 (Malvern) at 25°C with a setting of 20 × 2 µL injections. UCH37$^{C88A}$•RPN13 and Ub chains were all dialyzed into dialysis buffer (50 mM HEPES pH 7.4, 150 mM NaCl, and 500 µM TCEP). For all measurements, the syringe contained Ub or Ub chains at a concentration of 45–100 µM, and the cell contained UCH37$^{C88A}$•RPN13 at a concentration of 3–10 µM. The heats of dilution for diluting Ub chains into measurement buffer were subtracted from binding experiments before curve fitting. Manufactured supplied Origin software (OriginLab 7 SR4) was used to fit the data to a single-site binding model and to determine the stoichiometry (N), ΔH, ΔS, and the association constant $K_a$. The dissociation constant, $K_d$, was calculated from $K_a$.

## Ub-AMC and suc-LLVY-AMC assays

Ub-AMC and suc-LLVY-AMC were used to report on the Ub hydrolysis and peptidase activity, respectively, of UCH37 variants and replenished Ptsms. Assays were performed in black-well, clear-bottom 96-well plates (Corning). Mixtures containing UCH37•RPN13$^{DEUBAD}$, UCH37•RPN13 (20 nM), or Ptsms (1 µg) in their respective assay buffer (DUB buffer for UCH37•RPN13$^{DEUBAD}$ and UCH37•RPN13 and Ptsm reaction buffer for Ptsms) were incubated at 37°C for 20 min. The fluorogenic reagent (50 nM for Ub-AMC or 50 µM suc-LLVY-AMC) was then added to the appropriate wells, and hydrolysis was monitored continuously for 30 min at 37°C on a fluorescence plate reader (BioTek Synergy 2, $\lambda_{ex}$ = 360 nm, $\lambda_{em}$ = 460 nm).

## Ub chain pulldown assay with UCH37 variants

6×His-tagged UCH37 variants (5 nmol) were incubated with Ub chains composed of different linkages (50 pmol) and Ni-NTA resin (20 µL) in pulldown buffer (50 mM HEPES pH 7.5, 150 mM NaCl, 0.1% IGEPAL; 100 µL) for 2 hr at room temperature. The resin was washed 3× with pulldown buffer before captured Ub chains were eluted with 6× Laemmli loading buffer, separated using SDS-PAGE, and visualized by Coomassie staining.

## K6-linkage-specific affimer pulldown

The indicated cells were grown to about 60% confluency and treated with H$_2$O$_2$ for 1 hr. The medium was then replaced with medium containing 10 mM MG132, and the cells were grown for additional 4 hr. Emetine (25 µM) was then added to the medium, and the cells were harvested at indicated time points in lysis buffer (50 mM HEPES pH 7.5, 50 mM NaCl, 1× protease inhibitor cocktail). Cells were lysed by one freeze-thaw cycle followed by sonication. The resulting lysate was clarified by centrifugation at 20,000× g for 10 min. In parallel, the Halo-tagged K6-linkage-specific affimer was immobilized on Halo resin by incubating 500 µL of a 500 µM stock solution of Halo-tagged affimer with 300 µL of Halo beads in pulldown buffer (50 mM HEPES pH 7.5, 150 mM NaCl, 0.1% IGEPAL) for 2 hr at 4°C. The immobilized affimer was then washed 3× with wash buffer and incubated with 10 mg of cell lysate overnight at 4°C. The resin was then washed 5× with pulldown buffer prior to elution with 6× Laemmli loading buffer or treatment with different DUBs (e.g. 1 µM of OTUB1 or 2 µM UCH37•RPN13) for 1 hr at 37°C. The eluted proteins were resolved by SDS-PAGE, transferred onto a nitrocellulose membrane and immunoblotted with an αMfn2 antibody and αK48-linkage-specific Ub antibody.

## SEC-SAXS measurements and data processing

SEC-SAXS experiments were performed at Lawrence Berkeley lab (SIBYLS Beamline 12.3.1, Advanced Light Source). To ensure sample monodispersity, analytes were first separated by size-exclusion chromatography (SEC) using an Agilent 1260 series HPLC with a PROTEIN KW-802.5 analytical column (Shodex) at a flow rate of 0.5 ml/min. The column outlet was directly connected to the SAXS sample cell. Samples were then analyzed by SAXS as they emerge from the column with $\lambda$ = 1.03 Å incident light at a sample to detector distance of 1.5 m. The q-range sampled was from 0.013 to 0.5 Å$^{-1}$, where q = 4πsinθ/$\lambda$ and 2θ is the measured scattering angle. Three second exposures are collected for each frame over the course of 40 min during the SEC run. SEC-SAXS chromatographs are generated, and initial SAXS curves are analyzed using Scatter. Samples were analyzed at room temperature in 50 mM HEPES buffer pH 7.4, 50 mM NaCl, and 0.5 mM TCEP. Data were corrected for background scattering by subtracting the buffer curve from protein plus buffer curves. P(r) functions and $R_g$ values were determined from the scattering data using PRIMUS from the ATSAS software package.

## Hydrogen-deuterium exchange mass spectrometry

### Sample preparation and data collection

Complexes between Ub variants and UCH37·RPN13$^{DEUBAD}$ were generated by incubating UCH37·RP-N13$^{DEUBAD}$ (60 µM) with the Ub variant (180 µM) for 1 hr on ice in dilution buffer (50 mM HEPES, pH 7.4, 50 mM NaCl, and 1 mM TCEP). Next, 4 µL of protein complex was diluted into either 96 µL of H$_2$O buffer (20 mM sodium phosphate, pH 7.0) for reference mapping or 96 µL of D$_2$O buffer (20 mM sodium phosphate, pD 7.0 in D$_2$O) to initiate deuterium labeling. After 1, 10, 30, 60, and 120 min exposure to the deuterated buffer at 15°C, 50 µL labeling reaction was quenched by mixing with 50 µL quenching buffer (0.8 M Guanidine-HCl, 1% acetic acid, pH 2.2) at 4°C. Quenched samples were immediately injected into the HDX platform. Upon injection, samples were passed through an immobilized pepsin column (2.1 × 30 mm) (Waters Corp.) at 150 µL/min for inline digestion. Following digestion, the resulting peptides were desalted on a 1.8 µm C8 trap column (Waters Corp.) and separated on a 1.0 mm × 5 cm C18 column (Accuity) with linear gradient of 80% B (A is 0.1% formic acid [FA] in H$_2$O, and B is 0.1% FA in acetonitrile) over 12 min, and peptides were identified using ion mobility MS through a Synapt mass spectrometer (Waters Corp.).

### Peptide identification

Product ion spectra were acquired in data-dependent mode with a 1 s duty cycle, which means that the most abundant ions selected for the product ion analysis by higher-energy collisional dissociation between survey scans occurs once per second. The resulting MS data files were submitted to Protein-Lynx Global Server (PLGS) 3.0 software (Waters Corp.) to identify the peptic peptides in the undeuterated protein samples. Peptides included in the HDX analysis peptide set had a PLGS score greater than 7, and the MS spectra were verified by manual inspection.

### Data processing

Data from peptic peptides generated from PLGS were imported into DynamX 3.0 (Waters Corp.), with peptide quality thresholds of MS1 signal intensity greater than 5000, maximum mass error of 1 ppm, minimum products per amino acid greater than 0.3, and PLGS score greater than 7. Following the automated peptide search and ion detection, each spectrum was manually inspected to ensure the corresponding m/z and isotopic distributions at various charge states were properly assigned to the appropriate peptic peptide. The relative deuterium incorporation plot and overall deuterium uptake heat map for each peptic peptide were generated by DynamX 3.0. DynamX 3.0 calculated the relative deuterium incorporation by subtracting the weight-averaged centroid mass of the isotopic distribution of peptic peptides of undeuterated sample from that of samples exposed to deuterated buffer at various time points. All HDX-MS experiments were performed in triplicate, with single preparations of each purified protein complex. Since all differential HDX-MS experiments were conducted under the same conditions, the differences in the back exchange for each experiment would be negligible. Thus, the reported deuterium uptake levels are relative, neglecting the effect of the back exchange. The percent relative deuterium uptake was calculated by dividing the relative deuterium uptake of each peptic peptide by its theoretical maximum uptake. To generate heat maps of individual residues, the HDX data from all overlapping peptides were consolidated to individual amino acid values by averaging each residue. Briefly, for each residue, the deuterium incorporation values were calculated by dividing the total relative deuterium uptake of the peptic peptide by the peptide length. The differential deuterium uptake plot was generated by subtracting the relative deuterium uptake of peptic peptide of one complex by that of another. Statistical significance for the differential HDX data is determined by t-test for each time point, and a 98% confidence limit for the uncertainty of the mean relative deuterium uptake of ±0.6 Da was calculated as described. Differences in deuterium uptake between two states that exceed 0.6 Da is considered significant.

## Lentivirus packaging, infection, and cell line creation

Lentiviral packaging and target cell transductions were performed by the Cell Culture Core Facility at UMASS Amherst's Institute for Applied Life Sciences (Amherst, MA). Briefly, Lenti-X 293T cells (Takara Bio) were transfected with packaging and envelope plasmids psPAX2 and pMD2.G, respectively (gifts from Didier Trono, Addgene), and transfer plasmid pInducer21-UCH37-F117A, or pInducer21-UCH37-I216E. Lipofectamine2000 reagent (Invitrogen) was used to deliver DNA to the cells. Transfection

reagent was removed 6 hr after transfection and replaced with fresh EMEM (Lonza) supplemented with 10% FBS (Corning). Lentiviral supernatants were collected 18 hr later, passed through a 0.45 µm filter, and stored at −80°C.

HEK293FT UCH37$^{KO}$ cells were transduced with lentivirus in a six-well plate when cells were roughly 80% confluent. A threefold serial dilution of lentivirus was prepared in OptiMEM medium (Gibco) supplemented with 8 µg/mL polybrene (Millipore). Growth medium was removed from target cells and replaced with 2 mL dilution per well. Transduction plate was centrifuged at 2000 rpm for 2 hr at room temperature. Plate was incubated at 37°C/5% CO$_2$ for an additional 6 hr. Transduction medium was removed and replaced with fresh growth medium. Plates were incubated for an additional 40 hr before flow cytometry sorting.

## Emetine chase using GFP$^u$ reporter system

WT and UCH37$^{KO}$ GFP$^u$ cells were grown in 12-well plates until cells reached ~50–60% confluency and then treated with 36 µM emetine to shutoff translation for the indicated times. Cells were harvested in cold PBS containing 1 mM EDTA, followed by centrifuging at 1000× g for 2 min. Complementation experiments were performed by transfecting UCH37$^{KO}$ GFP$^u$ cells with plasmids expressing RFP under a constitutively expressed promoter and either UCH37$^{F117A}$ or UCH37$^{I216E}$ under a doxycycline inducible promoter. The transfected cells were grown in DMEM containing 0.1 µg/ml doxycycline for 48 hr in 12-well plates until ~40–50% confluency. Cells were then treated with 36 µM emetine for the indicated times, washed, harvested, and resuspended in FACS buffer (1 mM EDTA and 0.1% BSA in cold PBS) for flow cytometry. Data acquisition was performed on a BD LSR Fortessa X20 with 488 nm (530/30 band-pass collection filter), equipped with 488 nm and 561 nm lasers for excitation of GFP and RFP, respectively. Data analysis was performed using FlowJo version 10 (FlowJo, LLC). Statistical analysis was performed using Prism 8 and is represented as the mean fluorescence in BL1-A with standard deviation of three independent experiments. ***p<0.0005 (ANOVA two-way comparison).

## Pulse-chase experiments with L-azidohomoalanine labeling

UCH37$^{KO}$ cells expressing either UCH37$^{F117A}$ or UCH37$^{I216E}$ were seeded in six-well plates and grown to ~60% confluency. Cells were then washed once with PBS and incubated for 1 hr in methionine-free medium supplemented with or without 25 µM AHA. Cells were then either harvested or chased in complete DMEM supplemented with 2 mM excess methionine for indicated time points up to 4 hr. Negative controls were not treated with AHA and were harvested without chase. Carfilzomib controls were treated with 1 µM carfilzomib during both AHA incubation and 4 hr chase. Cells were harvested in cold PBS containing 1 mM EDTA and resuspended in RIPA buffer (10 mM Tris-HCl pH 7.5, 1 mM EDTA, 0.5% Triton X-100, 140 mM NaCl, 0.1% SDS, and 0.1% sodium deoxycholate supplemented with 1× protease inhibitor cocktail) before flash freezing on dry ice and storing at −80 °C. Pellets were lysed via 3× freeze-thaw cycles and sonicated for 3 min before centrifugation at 20,000× g for 10 min at 4°C. Total protein concentration was quantified using BCA assay. AHA-containing proteins were then labeled with 10 µM DBCO-Cy5 for 30 min at RT. Labeling was quenched with 6× Laemmli loading buffer without DTT and samples were separated by SDS-PAGE on 4–20% Mini-PROTEAN gels. Cy5 fluorescence and total protein staining using Sypro Ruby was measured on a Typhoon FLA 9500 (GE). For UCH37$^{KO}$ cells expressing either WT or UCH37$^{C88S}$, cells were seeded in medium containing 0.1 µg/mL doxycycline for 48 hr. Doxycycline remained present for the rescue cell lines after each medium change.

## $^{13}$C-Methyl ILV labeled protein expression

Ub monomers (Ub$^{D77}$ and Ub$^{K6R/K48R}$) were expressed in M9 minimal medium supplemented with [1,2,3,4,5,6,6-D7] D-glucose (2 g/L) in D$_2$O. Once cells reached an OD$_{600}$ ~1.0, [3-methyl-$^{13}$C, 3,4,4,4-D4] α-ketoisovaleric acid (120 mg/L) and [methyl-$^{13}$C, 3,3-D2] α-ketobutyric acid (70 mg/L) were added (*Tugarinov et al., 2006*). After an hour, 400 µM IPTG was added to induce expression. Purification of labeled Ub was performed as described above for unlabeled Ub.

## Synthesis of labeled Ub chains

The protocol for synthesizing Ub chains was modified slightly to generate labeled chains. Briefly, 1 mM Ub$^{D77}$, 1 mM Ub$^{K6R/K48R}$, 800 nM E1, and 4 µM UBE2R1 (K48-linkage-specific) were mixed in Ub

chain synthesis reaction buffer (10 mM ATP, 10 mM MgCl$_2$, 50 mM Tris-HCl pH 8.0, and 1 mM DTT) overnight at 37°C. Selective labeling of proximal and distal subunit of K48 di-Ub was achieved using either [$^{13}$C-methyl ILV] Ub$^{D77}$ or [$^{13}$C-Methyl ILV] Ub$^{K6R/K48R}$. Only one subunit was labeled at a time. Labeled K48 di-Ub was purified using the same procedure as unlabeled dimers.

## NMR spectroscopy

NMR spectra were collected on a Bruker Avance III HD spectrometer operating at 900 MHz ($^1$H) and equipped with a cryogenic probe. The temperature of the samples was regulated at 25°C throughout data collection. NMR samples were prepared in 99.9% D$_2$O, 20 mM potassium phosphate buffer (pH 6.8), and 0.1% (v/w) NaN$_3$. All NMR samples contained 30 μM mono-Ub or di-Ub [$^{13}$C-Methyl ILV]-labeled at the appropriate subunit. Titration experiments were performed with individually prepared samples at molar ratios of 1:0.5, 1:1, 1:1.5 for mono-Ub and 1:1, 1:1.5 for di-Ub. Two-dimensional (2D) $^1$H,$^{13}$C-SOFAST-HMQC spectra were collected with 1024 × 256 complex points in the direct $^1$H and indirect $^{13}$C dimensions, respectively. The power and pulse width for the selective 90 and 180° shaped pulses on the methyl $^1$Hs, PC9 and Reburp, respectively, were calculated on-the-fly within the pulse program from the 4.0 ppm bandwidth, while the pulse offset was set to 0.5 ppm. A recycle delay of 1 s was used, and 64 scans were accumulated for each FID for a total acquisition time of 10 hr. In addition, 2D $^1$H,$^{13}$C-ZZ-exchange experiments were performed on mono-Ub and K48 di-Ub labeled at either proximal or distal subunit. With the 1:1 UCH37$^{C88A}$•RPN13:mono-Ub complex, ZZ-exchange spectra were recorded using mixing times of 100, 200, 400, and 800 ms. With the proximal subunit of K48 di-Ub, ZZ-exchanged spectra were collected using mixing times of 100, 200, and 400 ms, while with the distal subunit labeled, mixing times of 100 and 200 ms were used. All ZZ-exchange spectra were recorded using 1024 × 128 complex points in the direct $^1$H and indirect $^{13}$C dimensions, respectively. The recycle delay was set to 1 s, and 512 scans were accumulated for each FID for a total acquisition time of approximately 2 days. All 2D spectra, SOFAST and ZZ-exchange, were recorded using a $^1$H spectral window of 16.3 ppm, while the $^1$H offset was set on water at 4.7 ppm. For the $^{13}$C dimension, the spectral window was set to 24.5 ppm and the offset was set to the middle of the methyl region at 16.6 ppm. All NMR data were processed with NMRPipe and analyzed using NMRFAM-Sparky.

## Molecular dynamics

MD simulations were performed on UCH37•RPN13$^{DEUBAD}$ using a homology model based on PDB ID: 4UEL. Missing N- and C-terminal residues of UCH37 were added using UCSF-Chimera (*Pettersen et al., 2004*), and homology modeling was done by the Swiss-Model webserver (*Schwede et al., 2003*) to fill in missing loops. Missing N- and C-terminal segments of RPN13$^{DEUBAD}$ were added from PDB ID:2KR0 and backbone dihedral angles adjusted to avoid clashes with UCH37. MD simulations were carried out with GROMACS-2018 using CHARMM36 force field (*Huang et al., 2017*). The complex was solvated in a TIP3P periodic dodecahedron box with a minimal distance of 9.5 Å to the box edge using counterions. The system was minimized using 5000 steps of steepest descent and heated to 310 K and equilibrated for 300 ps using an isothermal-isochoric ensemble (NVT). This was followed by 500 ps at the same temperature and 1 bar pressure using an isothermal isobaric ensemble (NPT). A total of eight independent simulations were performed for 50 ns each. To ensure proper geometries and bond lengths during the simulation, the LINCS algorithm was used with a 2 fs integration step and a cutoff of 12 Å for nonbonded interactions. Structures were clustered using the Jarvis Patrick algorithm with an RMSD cutoff of 3 Å, and the structure with the lowest RMSD to the average structure was used for further analysis.

## Docking simulations and fitting to SAXS data

Docking was performed using the HADDOCK2.2 webserver (*van Zundert et al., 2016*) to generate a structural model for UCH37•RPN13$^{DEUBAD}$ in complex with K48 di-Ub. Starting structures for UCH37•RPN13$^{DEUBAD}$ were selected from the best scoring multistate model to the SAXS data according to the FoXS webserver (*Schneidman-Duhovny et al., 2016*). The best scoring multistate model identified three conformations of UCH37•RPN13$^{DEUBAD}$, an additional two models were selected with lower scores but considerably different conformations to increase structural variability. Each conformation within the five-member ensemble was used in a separate docking simulation. Active residues on these structures were identified based the HDX-MS and crosslinking data. Residues F117 and F121 were

selected based on their effect on di-Ub binding and K154 due to the $K_m$ effect. Passive residues were selected based on HDX-MS. Residues that are protected upon K48 di-Ub binding on the CD and are solvent exposed were set as passive. N- and C-terminal segments of RPN13[DEUBAD] were left fully flexible during docking. Starting coordinates for K48 di-Ub were obtained from PDB ID: 1UBQ and used during multibody docking as proximal and distal subunits. Active residues for ubiquitin subunits were determined based on NMR titration experiments. Residues that belong to the I44 patch with identified exchange peaks were selected as active. Passive residues were identified within 4.5 Å cutoff of active residues and included other exchanging residues that were identified by NMR. Residues 72–74 and K48 of the proximal subunit were left fully flexible during all stages of docking. Unambiguous restraints were introduced between the ubiquitin subunits to preserve distances characteristic to isopeptide linkages (*Fushman and Walker, 2010*). Docking was performed using the standard HADDOCK protocol with minor adjustments. Rigid body docking was performed using 10,000 structures and the best 500 structures were selected for semiflexible refinement according to AIR energies. The resulting 500 structures were analyzed and clustered according to fraction of common contacts with a cutoff of 0.6 and minimal cluster size of 4. All structures obtained from clustering of MD simulations and docking were used to back calculate theoretical SAXS profiles and fitted against experimental data using the FoXS webserver. The calculation of SAXS profiles was performed with standard settings using an offset and background adjustment.

## Photocrosslinking of ubiquitin chains and UCH37

A photocrosslinker was installed on different subunits of K48 di-Ub and K6/K48 tri-Ub using a Ub variant in which A46 is replaced with Cys (A46C). The A46C bearing Ub chains were incubated with 500 µM MTS-alkynyldiazirine (Redbrick molecular) in 25 mM Tris, pH 8, 10% DMSO at room temperature for 30 min. Excess photocrosslinker was removed with Zeba spin column (Thermo Fisher scientific). The modified Ub chains (15 µM) were incubated with UCH37[C88A]•RPN13[DEUBAD] (45 µM) in crosslink reaction buffer (50 mM HEPES, pH 7.4, 50 mM NaCl) on ice for 1 hr. The mixture was then irradiated with a UV lamp (Omnicure, 320–500 nm) for 5 min on ice, followed by separation with SDS-PAGE and Coomassie staining. The gel slice containing crosslinked UCH37-Ub chain conjugate was cut out and reduced with 10 mM DTT (Ub peptide is released from the crosslinked peptide, but UCH37 peptide is still modified with the crosslinker) and then alkylated with 55 mM iodoacetamide. In-gel digestion was done with trypsin at 37°C overnight.

## Crosslinked peptide detection by LC-MS/MS

Extracted tryptic peptides were desalted and separated by a homemade fused silica capillary column (75 µm × 150 mm) packed with C-18 resin (120 Å, 1.9 µm, Dr. Maisch HPLC GmbH). Peptides were eluted by applying a 90 min gradient elution (solvent A: 0.1% FA in water/solvent B: 0.1% FA in acetonitrile) on the EASY-nLC 1000 nano-HPLC system (Thermo Fisher Scientific), coupled with the Orbitrap Fusion Tribrid mass spectrometer. Full MS scans were acquired at a resolution of 60,000 between 350 and 2000 m/z. Collision-induced dissociation (CID) was induced on the 10 most abundant ions per full MS scan using an isolation width of 20 ppm. Fragmented precursor ions were allowed for one repeated MS/MS analysis and then excluded for 15 s. LC-MS/MS data were searched against the manual database containing the UCH37[C88A]•RPN13[DEUBAD] and Ub sequence by using the Proteome Discoverer 2.4 with the SEQUEST HT search engine. Peptide's mass and MS/MS tolerance are set to 10 ppm and 0.6 Da. Methionine oxidation, N- terminal acetylation, cysteine carbamidomethylation, and a manually added photocrosslinker modification at any residue were used as a variable modification.

## Global proteomics analysis using tandem mass tag labeling

UCH37[KO] HEK293 FT cells transduced with WT UCH37, UCH37[F117A] and UCH37[I216E] were grown in DMEM supplemented with 10% FBS, 2 mM glutamine and 100 units/ml penicillin/streptomycin. Cells were plated in six-well plate and allowed to grow 12 hr before treating them with doxycycline to induce the expression of UCH37. After 12 hr doxycycline treatment, all cells were washed twice with cold PBS, and harvested with 1 mM EDTA in cold PBS. To induce the oxidative stress, cells were treated with 250 µM $H_2O_2$ for 1 hr prior to harvesting. Cells were then suspended in lysis buffer (8 M urea, 50 mM HEPES pH 7.4, 75 mM NaCl, and 1× protease inhibitor cocktail) and lysed by freeze-thaw cycles and sonication. Clarified lysate was then diluted in 100 mM TEAB buffer, followed by reduction

(20 mM TECP pH 7.5), alkylation (37.5 mM iodoacetamide), and precipitated by cold acetone overnight. Precipitated proteins were resuspended in 50 mM TEAB buffer and digested by trypsin overnight at 37°C. Then 20 µL of 20 ng/µL 10-plex TMT reagents (Thermo Scientific) dissolved in anhydrous acetonitrile were added to 50 µg of digested peptide samples. Following incubation at RT for 1 hr, the labeling reaction was quenched with hydroxylamine to a final concentration of 0.5% (v/v) for 15 min, and the equal volume of TMT labeled samples was pooled together. Pooled samples were dried using vacuum centrifuge and dried samples were dissolved in 10 mM $NH_4HCO_3$ pH 8.0 and fractionated by high pH reverse phase HPLC as previously described 39–40. Samples were fractionated into 96 fractions through an aeris peptide xb-c18 column (Phenomenex; 250 mm × 3.6 mm) with mobile phase A containing 5% acetonitrile and 10 mM $NH_4HCO_3$ in $H_2O$ (pH 8.5), and mobile phase B containing 90% acetonitrile and 10 mM $NH_4HCO_3$ in $H_2O$ (pH 8.5). The 96 resulting fractions were then pooled in a noncontinuous manner into eight fractions for subsequent mass spectrometry analysis. Samples were dried using vacuum centrifuge and resuspended in 0.1% FA in $H_2O$ for LC-MS/MS analysis. Mass spectrometry data were collected using an Orbitrap Fusion mass spectrometer (Thermo Fisher Scientific) coupled to a Proxeon EASY-nLC1200 liquid chromatography (LC) pump (Thermo Fisher Scientific). Peptides were separated on a 75 µm × 15 cm, nanoViper C18, 2 µm, 100 Å column (Thermo Fisher Scientific) with a gradient of 5% (0–10 min), 5–35% (10–150 min), 35–85% (150–160 min) solvent B (0.1% FA in acetonitrile) in solvent A (0.1% FA in $H_2O$) over a total 165 min run at 500 nL/min. 1–2 µg peptide from each fraction was injected onto the column for analysis. The scan sequence began with an MS1 spectrum (Orbitrap analysis; resolution 120,000; mass range 400–1400 m/z; automatic gain control [AGC] target 5 × 105; maximum injection time 100ms). MS2 analysis consisted of CID with 35% energy (AGC 5 × 103; isolation window 0.7 Th; maximum injection time 150 ms; activation time 10 ms). Monoisotopic peak assignment was used, and previously interrogated precursors were excluded using a dynamic window (150 s ± 7 ppm), and dependent scans were performed on a single charge state per precursor. Following acquisition of each MS2 spectrum, a synchronous-precursor-selection (SPS) MS3 scan was collected on the top 10 most intense ions in the MS2 spectrum. MS3 precursors were fragmented by high energy CID with 55% energy and analyzed using the Orbitrap (AGC 5 × 104; maximum injection time 150 ms, resolution was 10,000 at 2 Th isolation window).

## Acknowledgements

This work was funded by the NIH (RO1GM110543 to E.R.S.) and a NIH Chemistry and Biology Training Grant (T32GM008515 to J.D. and S.B.). The MS data described herein were acquired on either a Water Synapt G2Si or Orbitrap Fusion mass spectrometer in the MS Core Facility at UMass Amherst funded by NIH grant 1S10OD010645. This study also made use of the National Magnetic Resonance Facility at Madison, which is supported by NIH grant P41GM136463, old number P41GM103399 (NIGMS) and P41RR002301. Equipment was purchased with funds from the University of Wisconsin-Madison, the NIH P41GM103399, S10RR02781, S10RR08438, S10RR023438, S10RR025062, S10RR029220, the NSF (DMB-8415048, OIA-9977486, BIR-9214394), and the USDA. Preliminary NMR data were collected on a 600 MHz spectrometer in the NMR Facility at UMass Amherst.

## Additional information

### Competing interests

Eric Strieter: E.R.S. declares outside interest in Relay Therapeutics. The other authors declare that no competing interests exist.

### Funding

| Funder | Grant reference number | Author |
| --- | --- | --- |
| National Institutes of Health | R01GM110543 | Eric Strieter |

The funders had no role in study design, data collection and interpretation, or the decision to submit the work for publication.

## Author contributions

Jiale Du, Conceptualization, Data curation, Formal analysis, Investigation, Validation, Writing – original draft; Sandor Babik, Yanfeng Li, Conceptualization, Data curation, Formal analysis, Investigation, Validation; Kirandeep K Deol, Data curation, Formal analysis, Investigation, Validation; Stephen J Eyles, Formal analysis, Resources, Software; Jasna Fejzo, Data curation, Formal analysis, Resources; Marco Tonelli, Data curation, Formal analysis, Resources, Software; Eric Strieter, Conceptualization, Formal analysis, Funding acquisition, Project administration, Resources, Supervision, Validation, Writing – original draft, Writing – review and editing

## Author ORCIDs

Jiale Du http://orcid.org/0000-0003-1410-7020
Eric Strieter http://orcid.org/0000-0003-3447-3669

## Decision letter and Author response

Decision letter https://doi.org/10.7554/eLife.76100.sa1
Author response https://doi.org/10.7554/eLife.76100.sa2

## Additional files

### Supplementary files

• Transparent reporting form

• Source data 1. Tables of ITC, SEC-MALs, FP, Ub-AMC kinetics, SucLLVY, emetine chase, AHA pulse chase and TMT correlation analysis.

• Source data 2. Hydrogen Deuterium Exchange Mass Spectrometry analysis of UCH37 and RPN13$^{DEUBAD}$.

• Source data 3. Tandem Mass Tagging proteomics analysis.

• Source data 4. Uncropped gel images of Western blot analysis and gel based kinetic analysis.

• Source data 5. Full NMR spectra of mono-Ub and K48 di-Ub in presence and absence of UCH37.

### Data availability

All data generated or analysed during this study are included in the manuscript and supporting figures; Source Data files have been provided for all of the Figures.

The following dataset was generated:

| Author(s) | Year | Dataset title | Dataset URL | Database and Identifier |
|---|---|---|---|---|
| Du J, Babik S, Li Y, Deol KK, Fejzo J, Tonelli M, Strieter ER, Eyles SJ | 2021 | A Cryptic K48 Ubiquitin Chain Binding Site on UCH37 is Required for its Role in Proteasomal Degradation | https://data.mendeley.com//datasets/423f5cj79n/1 | Mendeley Data, 10.17632/423f5cj79n.1 |

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
