## [Editor Report]

This study identifies a previously unknown, non-canonical ubiquitin-binding site on the backside of the proteasome-associated deubiquitinase UCH37 that is responsible for the specific removal of Lys48-linked branches from ubiquitin chains. Using a broad array of biochemical and biophysical approaches, the authors characterize the ubiquitin binding modes and critical motifs of this new site, and investigate its effects on ubiquitin-dependent protein degradation by the 26S proteasome in vitro and in cells. These findings represent an important advance to our understanding of ubiquitin cleavage at the 26S proteasome and its role in regulating protein turnover by the ubiquitin-proteasome system.

---

## [Decision Letter]

**Decision letter after peer review:**

Thank you for submitting your article "A Cryptic K48 Ubiquitin Chain Binding Site on UCH37 is Required for its Role in Proteasomal Degradation" for consideration by *eLife*. Your article has been reviewed by 3 peer reviewers, including Andreas Martin as the Reviewing Editor and Reviewer #1, and the evaluation has been overseen by David Ron as the Senior Editor.

Essential revisions:

1) The reviewers primarily asked for clarifications, some additional information about experimental conditions, editing of the text, and possibly a better representation of the main findings, summarized in a model that illustrates the most likely pose of backside-bound branched ubiquitin and how it could reach the catalytic Cys.

2) To further elucidate the role of UCH37 for proteasomal substate degradation, distinguish between Ub-chain binding and debranching contributions, and assess the reason for the observed strong degradation inhibition when the backside binding site is mutated, it would be helpful to include a direct comparison between degradation data for backside-mutant and catalytically inactive UCH37, at least for the in vitro experiments.

*Reviewer #1 (Recommendations for the authors):*

This manuscript provides important new insights into UCH37-mediated ubiquitin binding and cleavage at the 26S proteasome, and is therefore well suited for publication in *eLife* after some concerns listed below have been addressed.

1) The HDX data in Figure S2 F and H show a stronger protection of the backside binding site by the branched Ub trimer if the front S1 site is occupied by a crosslinked mono-ubiquitin. The authors may consider commenting on this apparently better binding. Is this due to the differences in available binding sites or possibly indicating some sort of cooperativity?

2) When characterizing critical residues for backside Ub binding, the authors report ~ 20-fold reduced UCH37 debranching activity for the F117A and F121A mutants, and the table in Figure 6A lists kcat/Km values, but N/A for kcat and Km. How was kcat/Km determined, if the individual values are N/A?

(Also, the header of the table in Figure 6A should say "enzyme" not "substrate").

3) It is observed that proteasomes replenished with F117A or F121A mutant UCH37 fail to in-vitro degrade titin I27 or UBE2S-UBD model substrates carrying branched ubiquitin chains. This finding is intriguing, considering that the main proteasomal deubiquitinase Rpn11 cleaves off entire ubiquitin chains, and proteasomes for instance in budding yeast lack UCH37.

It would be worth testing proteasomes with catalytically inactive UCH37-C88A in these in vitro degradation experiments to distinguish whether UCH37's debranching activity is indeed required, or its backside binding may act as a receptor in orienting a substrate for degradation.

Based on their findings with cellular substrates, the authors hypothesize in the discussion that debranching may assist in the transfer of polyubiquitinated substrates from an effector protein to the proteasome, yet none such effector was present in the in vitro degradation experiments, and further investigating the role of UCH37 binding versus debranching may shine more light on that.

4) It is surprising that the turnover of AHA-labeled newly synthesized proteins (NSPs) in cells is completely inhibited by UCH37-F117A containing proteasomes (Figure 6H). Are indeed ALL NSPs expected to contain branched ubiquitin chains and depend on debranching for degradation?

In this regard it would also be interesting to know what approximate fraction of the GFPU model substrate contains branched chains.

Do the authors assume a possible dominant inhibitory effect of the UCH37 backside mutations, maybe because substrate-removed ubiquitin chains cannot be efficiently cleared or dissociate from the proteasome if they are still branched? Would the authors expect a difference between multiple- and single-turnover degradation of their polyubiquitinated titin or UBE2S model substrates?

Consistent with an inhibitory effect could also be that replenishing proteasomes in cells with the UCH37-F117A mutant apparently leads to 20% lower GFPU degradation than having no UCH37 at all (Figure 6G), and the authors may comment on that.

5) At odds with the observation that all NSPs are affected by the UCH37-F117A mutation appears to be that WT and UCH37-KO cells show only a minimal difference in total protein levels (Figure S7), and exposure to oxidative stress through treatment with H2O2 was necessary to see a significant difference suitable for the TMT-based proteomics experiments. In this context as well, the authors should consider testing the catalytically inactive UCH37-C88A mutant in comparison with the binding-deficient F117A mutant and the UCH37 KO.

Throughout the discussion, it is claimed that the turnover of various cellular proteins, including POLR2D and MFN2, is regulated by the K48 chain-specific binding and debranching activity of UCH37. However, only the binding aspect was directly assessed through the backside mutations, whereas the role of debranching activity in comparison was not tested, but implied.

6) The manuscript would benefit from a more specific model on how backside-bound branched ubiquitin chain may be oriented and get cleaved using the catalytic C88. Of course, the available data do currently not allow a detailed mechanistic view of debranching, but the discussion of binding modes and how a branched Ub chain may reach the catalytic active site could be improved to leave the reader with a more specific model than what is currently presented (e.g. in Figure 7I).

*Reviewer #2 (Recommendations for the authors):*

This manuscript is an important contribution to the field and worthy of publication in *eLife*. The comments below mostly offer suggestions to make the manuscript more accessible to a general audience.

Introduction: The authors should define 'S1' to make the manuscript more accessible to all readers.

Figure 1: In panel E (the SEC-MALS experiment), what is the peak that elutes at 19 minutes for sample UCH37(CD)C88A:K48 triUb?

On p. 6, it's stated that Rpn13-DEUBAD addition enables investigation of a wider range of Ub interactions, but it's not clear why this would be the case.

The use of the word 'extant' at line 169 p. 6 may need to be checked and as well in a later appearance.

Figure 2: residues in the surface diagrams should be labeled, especially on the rotated structures. It might also be worth labeling where the S1 site is.

The acronym CL for cross-over loop on p. 6 is not defined, nor introduced.

In Figure 3 would be more useful if it also included the location of the catalytic cysteine (C88), the canonical S1 site labeled, and more residue numbers. This same recommendation applies to Figure S3D. The table in Figure S3C would be more useful if it was color coded (or indicated in some other way) to show peptides corresponding to the front versus back binding site.

p. 9, line 270: the relative differences for peptides 183-206 and 211-217 should be explicitly stated, possibly better represented by normalizing and plotting the data in Figure S3C, and/or Figure 3 could be referenced here.

Figure 4: Pose 3 seems to be unsupported by the data shown in Figure 3. How does the SAXS data fit to the UCH37:Rpn13-DEUBAD structure without ubiquitin? This could be used as an indicator of the sensitivity of this method to Ub binding in this context. Perhaps pose 3 should be tested further if included as a main figure by adding di-Ub to Rpn13-DEUBAD bound to the ULD. To avoid confusion, the word 'model' should probably be included with 'Pose'. Similarly, D-H could be confused as definitive structures. To validate the models further, amino acid charge swapping could done for panels F, G and H. This additional experimental data may not be necessary if it's clearer that the structures shown are only models.

Figure 5: The molar ratio and concentrations should be included for the NMR data. What are the small peaks that appear in some but not all spectra – for example in the complex of B (green), free state for D and monoUb of the F117A complex of F? Why do they vary in intensity? Also why does I44 at the apparent free state position seem to have a split resonance peak shape in the complexed F117A sample? Perhaps the free monoUb spectrum should be included for comparison.

The HDX data seem to indicate that monoUb binds to the front side and the F117A mutation is in the backside. However monoUb binding is apparently affected by NMR in the F117A mutation (Figure 5F). This combination of data seems to suggest that the F117A mutation causes allosteric effects at the frontside as well as loss of binding to the backside. If true this may affect interpretation of the data in Figure 6 and 7.

Figure 7I needs labeling. What is in orange exactly? Is the model trying to show a Ub in the chain binding to the DEUBAD in the backside binding mode? This figure amplifies concerns regarding Figure 4. There is NMR evidence that Rpn13-DEUBAD does not bind to monoUb albeit without UCH37 present (DOI 10.1016/j.molcel.2010.04.019, Figure S4A).

---

## [Author Response]

Reviewer #1 (Recommendations for the authors):This manuscript provides important new insights into UCH37-mediated ubiquitin binding and cleavage at the 26S proteasome, and is therefore well suited for publication in eLife after some concerns listed below have been addressed.1) The HDX data in Figure S2 F and H show a stronger protection of the backside binding site by the branched Ub trimer if the front S1 site is occupied by a crosslinked mono-ubiquitin. The authors may consider commenting on this apparently better binding. Is this due to the differences in available binding sites or possibly indicating some sort of cooperativity?

The measured K_d_s for K6/K48 tri-Ub binding to either free or Ub-bound UCH37 are quite similar (see Deol et al. Mol Cell 2020 and data now shown in Figure 2 – Supplementary Figure 2), suggesting there is little cooperativity when Ub occupies the S1 site. Moreover, the addition of K48 di-Ub also results in stronger protection on the backside when the S1 site is blocked (compare Figure 2C and 2D) and the measured K_d_s for binding free and Ub-bound forms are nearly the same.

We surmise that when the S1 site is free, a Ub chain can sample either the front or backside of UCH37. However, when the S1 site is blocked, binding only occurs on the backside. This shift in the population of Ub chain bound UCH37 could lead to stronger protection on the backside. We have added a few sentences in the text to clarify the differences we observe in the uptake plots:

“It appears that when Ub is conjugated to UCH37, binding of either K48 di-Ub or K6/K48 tri-Ub affords stronger protection of the α5–6 motif (Figures 2D, S2H). This could be due to an allosteric effect resulting in enhanced binding or a situation in which K48 chains only interact with the backside because sampling of both faces is shutdown. Binding data support the latter by showing that Ub conjugation does not improve the affinity toward K48 di-Ub or K6/K48 tri-Ub (Figure 2 – Supplementary Figure 2).” Pages 7-8, Lines 216-222.

2) When characterizing critical residues for backside Ub binding, the authors report ~ 20-fold reduced UCH37 debranching activity for the F117A and F121A mutants, and the table in Figure 6A lists kcat/Km values, but N/A for kcat and Km. How was kcat/Km determined, if the individual values are N/A?(Also, the header of the table in Figure 6A should say "enzyme" not "substrate").

We thank the reviewer for noticing an oversight on our part. In general, the reported values were obtained by fitting the kinetic data to the Michaelis-Menten equation. The resulting *k*_cat_ and *K*_m_ parameters were then used to calculate *k*_cat_/*K*_m_. This method can be applied to data in which saturation is achieved, e.g., with WT UCH37 and I216E. However, saturation was not achieved with the L181A, F117A or the F121A variant. Thus, the values obtained for *k*_cat_ and *K*_m_ using a non-linear equation to fit the data are inaccurate. For L181A, F117A and F121A, the linear equation, rate=*k*_cat_/*K*_m_ [K6/K48 tri-Ub] [UCH37], can be used to fit the data with R^2^ values ≥ 0.9. The corrected *k*_cat_/*K*_m_ values are now reported in the table in Figure 6A. We have also indicated which equation was used to fit each data series in the Figure caption.

3) It is observed that proteasomes replenished with F117A or F121A mutant UCH37 fail to in-vitro degrade titin I27 or UBE2S-UBD model substrates carrying branched ubiquitin chains. This finding is intriguing, considering that the main proteasomal deubiquitinase Rpn11 cleaves off entire ubiquitin chains, and proteasomes for instance in budding yeast lack UCH37.It would be worth testing proteasomes with catalytically inactive UCH37-C88A in these in vitro degradation experiments to distinguish whether UCH37's debranching activity is indeed required, or its backside binding may act as a receptor in orienting a substrate for degradation.

We apologize for any confusion. In this initial submission, we show that proteasomes replenished with inactive UCH37 C88A (rC88A) are unable to promote the degradation of polyubiquitinated titan and UBE2S-UBD (Figure 6D). We also show similar results in our 2020 Molecular Cell paper. The results mirror those using proteasomes replenished with either F117A or F121A (shown in the same gel as rC88A; Figure 6D). Thus, the failure to degrade proteins by UCH37 C88A- or F117A/F121A-replenished proteasomes suggests the lack of activity with the latter is due to the inability to debranch and not just the loss of K48 chain binding. We realize the significance of these findings was not clearly conveyed in the main text. We have now added a sentence clarifying our findings:

“In the presence of K6/K48 and K48/K63 HMW Ub chains, WT- and I216E-replenished (rWT and rI216E) Ptsms generate smaller Ub conjugates, but the results with F117A- and F121A-replenished (rF117A and rF121A) Ptsms essentially mirror those with the inactive rC88A variant (Figure 6C). Degradation is also affected by F117A and F121A. Using K11/K48-polyUb-UBE2S-UBD and K48/K63-polyUb-titin-I27^V15P^ as substrates, a decrease in polyubiquitinated species and a concomitant increase in Ptsm-derived peptides is observed with rWT and rI216E Ptsms but not with rF117A, rF121A, and rC88A Ptsms (Figure 6D).

Because the rC88A Ptsms retain the ability to bind K48 chains, these results imply that the loss of chain debranching, not just chain binding, leads to the failure to degrade branched polyubiquitinated proteins.” Page 14, lines 410-418.

Based on their findings with cellular substrates, the authors hypothesize in the discussion that debranching may assist in the transfer of polyubiquitinated substrates from an effector protein to the proteasome, yet none such effector was present in the in vitro degradation experiments, and further investigating the role of UCH37 binding versus debranching may shine more light on that.

Effector proteins are indeed absent from our in vitro degradation assays. In the revised Discussion, we state that due to the absence of effector proteins in our degradation assays, we favor a model in which chain debranching by UCH37 facilitates the clearance of chains removed by RPN11 after each round of degradation. Thus, UCH37 prevents product inhibition.

“In our in vitro degradation assays, however, these shuttling factors are largely absent.

The model we favor is one in which debranching facilitates clearance of chains removed by the intrinsic proteasomal DUB RPN11 after each round of degradation. Unfolding and translocating Ub chains through the pore of the proteasome presents a major roadblock for the degradation process (Worden et al., 2017). Co-translocational removal of Ub modifications by the Zn^2+^-dependent JAMM/MPN DUB RPN11 (Verma et al., 2002; Yao and Cohen, 2002), which resides directly above the entrance to the pore of the 19S AAA+ ATPases (Dambacher et al., 2016), alleviates this issue. However, the Ub chains that are cleaved en bloc by RPN11 can remain bound to the proteasome and inhibit additional rounds of degradation. This scenario could be more acute with branched chains compared to their unbranched counterparts considering the 19S subunit RPN1 strongly prefers binding K11/K48 branched chains (Fushman Structure paper). Debranching by UCH37 could then attenuate Ub chain binding, freeing the Ub receptors of the proteasome for another round of degradation.” Page 20, lines 600-612.

4) It is surprising that the turnover of AHA-labeled newly synthesized proteins (NSPs) in cells is completely inhibited by UCH37-F117A containing proteasomes (Figure 6H). Are indeed ALL NSPs expected to contain branched ubiquitin chains and depend on debranching for degradation?In this regard it would also be interesting to know what approximate fraction of the GFPU model substrate contains branched chains.

We do not have proof that all NSPs with branched chain modification are dependent on debranching for degradation, but our data using debranching-deficient UCH37 suggests that the degradation of majority of NSPs is dependent on UCH37.

That said, this is a very interesting question. We are quite keen on understanding to what extent proteins are modified with branched chains. We are currently using various enrichment strategies along with middle-down MS to assess the degree of chain branching. Our findings will be reported in a separate manuscript in the near future.

To more directly address the reviewer’s point, the data in Figure 7H would seem to suggest that the UCH37 substrate MFN2 is primarily modified with branched chains. The addition of OTUB1 and UCH37 to polyubiquitinated MFN2 leads to nearly complete dismantling of the chains (lanes 2-3 and 5-6 in the MFN2 blot on top).

Do the authors assume a possible dominant inhibitory effect of the UCH37 backside mutations, maybe because substrate-removed ubiquitin chains cannot be efficiently cleared or dissociate from the proteasome if they are still branched? Would the authors expect a difference between multiple- and single-turnover degradation of their polyubiquitinated titin or UBE2S model substrates?

Our hypothesis is that UCH37 prevents product inhibition after each round of degradation by debranching substrate-removed ubiquitin chains and thus facilitating dissociation (see lines 594-609 in the revised manuscript). According to this model, we would predict that UCH37 would have little impact on the degradation process under single-turnover conditions. Under multi-turnover conditions, such as those reported in this manuscript, however, there would be a dependence on UCH37’s ability to debranch chains. We are currently testing this hypothesis and will report the findings in another manuscript.

Consistent with an inhibitory effect could also be that replenishing proteasomes in cells with the UCH37-F117A mutant apparently leads to 20% lower GFPU degradation than having no UCH37 at all (Figure 6G), and the authors may comment on that.

We agree with the reviewer and added the following sentences.

“Of note, the degradation of GFP^u^ is lower in cells expressing UCH37^F117A^ compared to the null background. This could be due to an inhibitory effect caused by the inability to efficiently clear Ptsmbound Ub chains that are removed from the substrate during translocation into the proteolytic chamber (see Discussion).” Pages 14-15, lines 431-434.

5) At odds with the observation that all NSPs are affected by the UCH37-F117A mutation appears to be that WT and UCH37-KO cells show only a minimal difference in total protein levels (Figure S7), and exposure to oxidative stress through treatment with H2O2 was necessary to see a significant difference suitable for the TMT-based proteomics experiments. In this context as well, the authors should consider testing the catalytically inactive UCH37-C88A mutant in comparison with the binding-deficient F117A mutant and the UCH37 KO.

As shown in Figure 6I, the majority of NSPs are turned over rather quickly. Most short-lived proteins are in low abundance (Li et al. Mol Cell 2021) and thus difficult to detect under steady-state conditions such as those in Figure 7 – Supplementary Figure 1. Thus, we think there is little overlap between the NSPs in Figure 6I and the proteins in Figure 7 – Supplementary Figure 1.

Throughout the discussion, it is claimed that the turnover of various cellular proteins, including POLR2D and MFN2, is regulated by the K48 chain-specific binding and debranching activity of UCH37. However, only the binding aspect was directly assessed through the backside mutations, whereas the role of debranching activity in comparison was not tested, but implied.

In in vitro degradation experiments we found that proteasomes replenished with either inactive UCH37 (C88A) or backside mutants fail to degrade two different substrates with different branched chains (Figure 6D). If the inability to bind chains with the backside mutant was the sole reason for the impaired degradation, then we would expect to observe degradation with the binding competent C88A mutant. However, C88A also shuts down the degradation of branched chain-modified proteins (Figure 6D). Based on these results, we thus assume that the turnover of proteins like POLR2D and MFN2 is regulated by the K48 chain-specific binding and debranching activity of UCH37.

6) The manuscript would benefit from a more specific model on how backside-bound branched ubiquitin chain may be oriented and get cleaved using the catalytic C88. Of course, the available data do currently not allow a detailed mechanistic view of debranching, but the discussion of binding modes and how a branched Ub chain may reach the catalytic active site could be improved to leave the reader with a more specific model than what is currently presented (e.g. in Figure 7I).

We added Figure 4I to show how the isopeptide bond is positioned relative to the catalytic cysteine in two different docking models. We also added Figure 8 to better represent the overall findings of our manuscript.

Reviewer #2 (Recommendations for the authors):This manuscript is an important contribution to the field and worthy of publication in eLife. The comments below mostly offer suggestions to make the manuscript more accessible to a general audience.Introduction: The authors should define 'S1' to make the manuscript more accessible to all readers.

The introduction has been revised to better define the S1 site and describe how chain debranching could occur by placing the K48 distal ubiquitin in the S1 site.

“Based on known structures of monoUb bound to UCH37 (Sahtoe et al., 2015a; VanderLinden et al., 2015), the hypothesis is that both the C-terminal hydrolase and debranching activities utilize the same S1 site during catalysis. The S1 site is composed of a series of residues that form a rim leading into the active site (Figure 1A). The DEUBAD domain of RPN13 further promotes Ub binding by properly positioning both the flexible active site crossover loop (CL), which is a characteristic feature of all UCHs (Johnston et al., 1999; Popp et al., 2009), and the C-terminal helical region of UCH37. One would therefore speculate that during chain debranching, the S1 site would be occupied by the K48-linked distal Ub, as this subunit is the one removed from the branched chain (Figure 1B). In this orientation, the other two Ub subunits at the branchpoint would also be expected to engage UCH37; however, the precise mechanism by which branched chains are selectively processed by UCH37 remains unknown. A region that has received little attention in any of the UCHs is the face located on the opposite side of the CL relative to the S1 site. We thought that by defining how K48 chains interact with UCH37, we could begin to dissect the impact of K48 chain binding and debranching on a proteome-wide level.” Page 4, lines 109-124.

Figure 1: In panel E (the SEC-MALS experiment), what is the peak that elutes at 19 minutes for sample UCH37(CD)C88A:K48 triUb?

The UV peak at 19 minutes does not have any light scattering data, therefore no molecular weight information can be extracted from the peak. We add clarification to this in the figure caption for Figure 1E. Addition of DEUBAD releases UCH37 from autoinhibition, thus enhancing its binding affinity towards both mono-Ub and Ub chains. We added a statement in the manuscript to clarify this point:

“* denotes a UV peak without corresponding light scattering data.” Page 24, lines 693-694.

On p. 6, it's stated that Rpn13-DEUBAD addition enables investigation of a wider range of Ub interactions, but it's not clear why this would be the case.

Addition of DEUBAD releases UCH37 from autoinhibition, thus enhancing its binding affinity towards both mono-Ub and Ub chains. We added a statement in the manuscript to clarify this point:

“Like the CD, the UCH37•RPN13^DEUBAD^ complex binds K48 chains with 1:1 stoichiometry (Figures S1F-H), but more importantly, the addition of RPN13^DEUBAD^ releases UCH37 from an autoinhibited state thereby enhancing the binding affinity toward mono-Ub (Hamazaki et al., 2006; Qiu et al., 2006; Sahtoe et al., 2015a; VanderLinden et al., 2015; Yao et al., 2006). This enables the investigation of a wide range of interactions from mono-Ub to branched trimers, which is necessary for uncovering cryptic Ub binding sites outside of the canonical S1 (cS1) site.” Page 6, lines 165-172.

The use of the word 'extant' at line 169 p. 6 may need to be checked and as well in a later appearance.

We replaced the word ‘extant’ with ‘existing’ and ‘previously reported’.

Figure 2: residues in the surface diagrams should be labeled, especially on the rotated structures. It might also be worth labeling where the S1 site is.

Figure 2 has been revised according to reviewer’s suggestion.

The acronym CL for cross-over loop on p. 6 is not defined, nor introduced.

The CL has now been introduced in the Introduction. Page 4, lines 112-115.

In Figure 3 would be more useful if it also included the location of the catalytic cysteine (C88), the canonical S1 site labeled, and more residue numbers. This same recommendation applies to Figure S3D. The table in Figure S3C would be more useful if it was color coded (or indicated in some other way) to show peptides corresponding to the front versus back binding site.

Figure 3 and Figure 3 – Supplementary Figure 1 have been revised according to the reviewer’s suggestion.

p. 9, line 270: the relative differences for peptides 183-206 and 211-217 should be explicitly stated, possibly better represented by normalizing and plotting the data in Figure S3C, and/or Figure 3 could be referenced here.

In the revised manuscript, we normalized the relative abundance of crosslinked peptides within individual Ub subunits. This improves visualization of the crosslinked data in Figure 3 and Figure 3 – Supplementary Figure 1.

Figure 4: Pose 3 seems to be unsupported by the data shown in Figure 3. Perhaps pose 3 should be tested further if included as a main figure by adding di-Ub to Rpn13-DEUBAD bound to the ULD.

Pose 3 is only supported by the HDX (DEUBAD) and SAXS data. Crosslinked peptides corresponding to the N-terminal portion of the crossover loop and the DEUBAD domain were not detected, but this could be due to the position of the diazirine on individual subunits. Thus, we do not want to discount pose 3. That said, we moved pose 3 the supplemental since we focus on poses 1 and 2 in the main text.

We added a sentence to the text to clarify where pose 3 comes from:

“In the third pose, one of the Ub subunits can be seen interacting with the CL and the DEUBAD domain, which is consistent with the HDX data (Figures 2J-K)”. Page 11, lines 320-322.

How does the SAXS data fit to the UCH37:Rpn13-DEUBAD structure without ubiquitin? This could be used as an indicator of the sensitivity of this method to Ub binding in this context.

A direct comparison of the SAXS data for UCH37•RPN13DEUBAD and K48 di-

Ub:UCH37•RPN13DEUBAD is shown in Figure 4 – Supplementary Figure 2. These data show that the presence of K48 di-Ub induces significant changes in the scattering curve.

To avoid confusion, the word 'model' should probably be included with 'Pose'. Similarly, D-H could be confused as definitive structures. To validate the models further, amino acid charge swapping could done for panels F, G and H. This additional experimental data may not be necessary if it's clearer that the structures shown are only models.

In Figure 4, we made it more clear that the poses are from docking models.

Figure 5: The molar ratio and concentrations should be included for the NMR data. What are the small peaks that appear in some but not all spectra – for example in the complex of B (green), free state for D and monoUb of the F117A complex of F? Why do they vary in intensity?

The molar ratios have been added to Figure 5 and the concentrations have been added to the figure caption. With regards to the small peaks, we do not know what these resonances correspond to or why they vary in intensity. However, we do observe those same resonances in the absence of UCH37•RPN13^DEUBAD^, indicating that these small peaks do not correspond to a new bound state of the Ub chains.

Also why does I44 at the apparent free state position seem to have a split resonance peak shape in the complexed F117A sample? Perhaps the free monoUb spectrum should be included for comparison.

We think I44 splitting is due to the presence of two distinct rotamers. Similar splitting is observed for I44 of the proximal Ub subunit in the unbound form of K48 di-Ub (see Figure 4B).

The HDX data seem to indicate that monoUb binds to the front side and the F117A mutation is in the backside. However monoUb binding is apparently affected by NMR in the F117A mutation (Figure 5F). This combination of data seems to suggest that the F117A mutation causes allosteric effects at the frontside as well as loss of binding to the backside. If true this may affect interpretation of the data in Figure 6 and 7.

Our HDX and ITC data show that monoUb can actually bind both the front and back of UCH37. We surmise that the binding of monoUb to both front and back is reflected in the NMR spectrum of monoUb bound to WT (aka C88A) UCH37•RPN13^DEUBAD^. In the spectrum with the F117A mutant, we are now only looking at binding to one of those sites, namely the frontside. The NMR spectrum of I216E bound to mono-Ub also supports this conclusion (see Author response image 1). In the I216E complex, we observe resonances corresponding to the bound and unbound states of monoUb. However, in the spectrum with the C88A complex, we only observe the bound peaks. It is important to mention that these data do not rule out the possibility of an allosteric effect due to the F117A or I216E mutation.

**Author response image 1. sa2fig1:** 

Figure 7I needs labeling. What is in orange exactly? Is the model trying to show a Ub in the chain binding to the DEUBAD in the backside binding mode? This figure amplifies concerns regarding Figure 4. There is NMR evidence that Rpn13-DEUBAD does not bind to monoUb albeit without UCH37 present (DOI 10.1016/j.molcel.2010.04.019, Figure S4A).

We replaced the model in Figure 7I with a new figure, Figure 8, which provides a more detailed explanation of our overall findings.